# Contractive MASO-Generalized Predictors for Stable Latent-Space Learning in JEPA

## Abstract

Joint Embedding Predictive Architectures (JEPAs) learn representations by predicting latent target embeddings from contextual views, but their predictors are typically shallow feed-forward networks with limited control over multi-step dynamics and stability. We introduce *Learnable Iterated Function Systems* (LIFS), a recursive and contractive latent operator that replaces the standard JEPA predictor with a mixture of affine maps applied over multiple refinement steps. The mixture weights are conditioned on the context embedding, enabling input-adaptive geometric structure while preserving the original JEPA objective and encoder architecture. LIFS can be viewed as a dynamical extension of *Max-Affine Spline Operators* (MASOs): instead of selecting a single affine branch, the latent state evolves through a sequence of spectrally-controlled affine transformations, yielding trajectory-dependent partitions of the latent space and explicit contraction guarantees. We establish sufficient conditions under which LIFS defines a Banach contraction and analyze its training behavior through Lyapunov-style stability arguments. Empirically, integrating LIFS into JEPA leads to smoother training dynamics and consistent improvements in predictive alignment and linear-probe accuracy, particularly for ViT encoders, without increasing model capacity. Overall, LIFS provides a principled and modular way to endow JEPA predictors with stable multi-step latent refinement, bridging predictive self-supervision, MASO theory, and contraction-based dynamical systems.

## 1 Introduction

Self-supervised learning (SSL) (Misra & van der Maaten, 2020; Balestriero et al., 2023; Cabannes et al., 2023; Simon et al., 2023) has advanced rapidly through latent-prediction objectives that avoid pixel-level reconstruction. Methods such as VICReg (Bardes et al., 2022), SimMIM (Xie et al., 2022), BYOL (Grill et al., 2020), iBOT (Zhou et al., 2022), Data2Vec (Baevski et al., 2022; 2023), and Joint Embedding Predictive Architectures (JEPAs) (LeCun, 2022; Balestriero & LeCun, 2025) learn representations by predicting latent embeddings of unseen views using a predictor network, without contrastive negatives. JEPA in particular avoids generative reconstruction while preserving semantic structure by operating entirely in latent space Schmidhuber & Prelinger (1993), and recent variants such as I-JEPA (Assran et al., 2023b;a;c; Bardes et al., 2024) demonstrate strong performance even when context and target views do not overlap.

Despite this progress, the design of the predictor itself has received comparatively little attention. In most JEPA implementations, the predictor is a shallow feed-forward or residual MLP applied once to the context embedding. While effective for short-horizon prediction, such predictors lack explicit control over multi-step refinement, stability, and latent-space dynamics—properties that become increasingly important in non-overlapping or long-horizon settings. To address this limitation, we revisit the geometry of the JEPA predictor and propose *Learnable Iterated Function Systems (LIFS)*, a recursive and contractive latent operator that replaces the standard one-step predictor. LIFS is inspired by the MASO/spline view of deep networks (Balestriero et al., 2018; Balestriero & Baraniuk, 2020), which shows that networks with piecewise-affine nonlinearities can be expressed as *Max-Affine Spline Operators (MASOs)*(sec. 2.3), and by Iterated Function Systems (IFS), built from repeated contractive transformations (Hutchinson, 1981; Barnsley, 1988). Instead of selecting a single affine branch, LIFS applies a mixture of affine maps over multiple refinement steps, by learning both the transformations and their mixture weights end-to-end, with input-dependent gating and explicit spectral control (Yoshida & Miyato, 2017; Miyato et al., 2018; Sedghi et al., 2019). This yields a stable multi-step latent evolution process that enhances expressivity while preserving contraction (cf. Lemma 4.1).

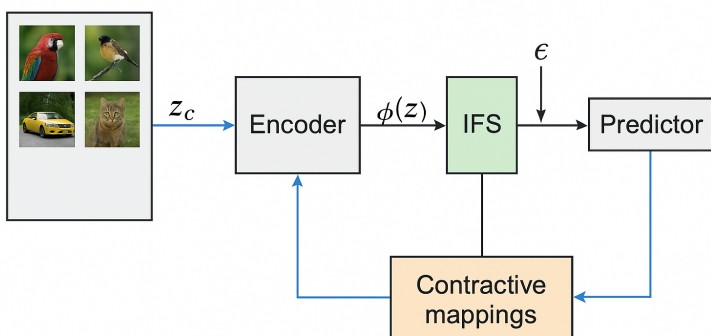

Figure 1: Architectural Integration: IFS-Augmented JEPA

Our formulation is recursive rather than generative: repeated contractive updates refine the latent state across iterations, producing a trajectory of intermediate predictions. This multi-step refinement captures structure across latent-space scales without modifying the encoder, the JEPA objective, or the masking strategy. Importantly, LIFS is a drop-in replacement for the predictor, making it easy to integrate into existing JEPA pipelines (cf. Figure 1).

From a theoretical perspective, we show that LIFS defines a Banach contraction under mild spectral constraints, ensuring a unique fixed point (Bai et al., 2019) and stable convergence. We further analyze its training behavior using Lyapunov-style arguments (cf. Sec. 6) and show that the operator induces *contractive gradient transport*, stabilizing encoder updates and preventing gradient explosion (cf. Sec A.0.7).

Empirically, integrating LIFS into JEPA with both ResNet (He et al., 2016) and ViT (Dosovitskiy et al., 2021) backbones yields smoother training dynamics, bounded spectral norms, and consistent improvements in predictive alignment and linear-probe accuracy, with the largest gains observed for ViT encoders. These results suggest that structuring the predictor as a contractive recursive operator provides a principled and modular path toward more stable and robust self-supervised representation learning.

## 2 Related Work

Self-supervised learning has advanced along several methodological directions (Caron et al., 2018; Zbontar et al., 2021; Caron et al., 2021; Chen et al., 2021), including contrastive learning (Chen et al., 2020; Wang & Isola, 2020), non-contrastive bootstrap methods (van den Oord et al., 2018; Grill et al., 2020; Tian et al., 2021), masked prediction architectures (He et al., 2022), and more recently Joint Embedding Predictive Architectures (JEPAs) (Assran et al., 2023c;a). Our work builds on this last family by revisiting the geometry of the JEPA predictor and introducing a recursive, contractive alternative.

### 2.1 Self-supervised predictive learning.

Early contrastive frameworks such as SimCLR (Chen et al., 2020) rely on explicit negative samples to avoid representational collapse. Non-contrastive or *bootstrap* methods such as BYOL (Grill et al., 2020) and VICReg (Bardes et al., 2022) eliminate negative samples through architectural asymmetry, feature standardization, and covariance constraints. These models improve training stability but their latent transformations remain fundamentally feed-forward, lacking explicit mechanisms to encode multi-scale or hierarchical geometric structure.

Mask-based prediction methods such as MAE (He et al., 2022; El-Nouby et al., 2023) and teacher–student latent prediction frameworks such as Data2Vec (Baevski et al., 2022; 2023) introduce cross-view prediction but still rely on shallow, single-step predictors. In contrast, our approach introduces an explicit *recursive contractive operator* that refines latent predictions over multiple steps while preserving JEPA's training objective.

## 2.2 Joint Embedding Predictive Architectures

JEPAs formulate self-supervised learning as latent-space prediction between non-overlapping context and target blocks, avoiding pixel reconstruction and contrastive negatives. I-JEPA (Assran et al., 2023c;a;b; Bardes et al., 2024) demonstrates that strong representations can be learned even when context and target views do not overlap. However, the predictor in I-JEPA is deliberately simple and applied only once, which limits its ability to capture:

- multi-step latent refinement,

- compositional or recursive structure,

- stable long-horizon latent dynamics.

Recent extensions explore architectural variations (Carr et al., 2024; Mo et al., 2024; Bardes et al., 2024; Wang et al., 2025), but none introduce an explicit *recursive geometric operator*. Our work fills this gap by replacing the one-step predictor with a multi-step *Learnable Iterated Function System (LIFS)* (cf. Figure 2), which provides controlled contraction, richer latent dynamics, and a principled dynamical-systems interpretation of JEPA prediction.

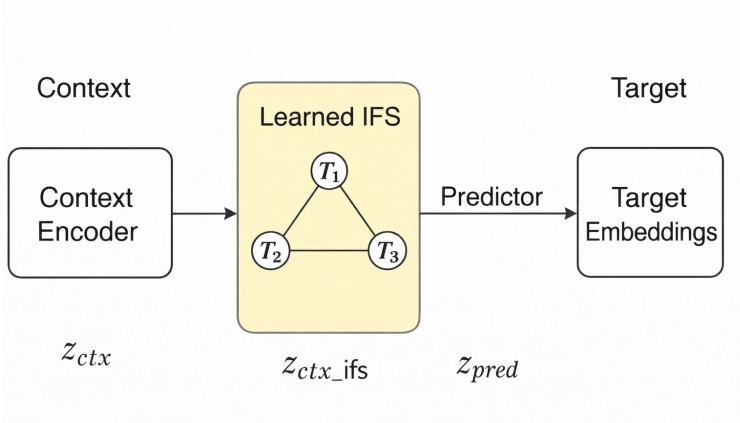

Figure 2: JEPA + Learnable IFS (LIFS) Training Pipeline. The Learnable IFS refines context embeddings via iterative and contractive latent transformations, enabling multi-scale structure modeling prior to predictive alignment.

## 2.3 Relation to MASO Spline Theory

LIFS can be viewed as a learnable, recursive generalization of the *Max-Affine Spline Operator (MASO)* framework (Balestriero et al., 2018; Balestriero & Baraniuk, 2020), which expresses deep networks with piecewise-affine nonlinearities as compositions of affine templates selected by region assignments. In LIFS, each affine map $T_k(z) = A_k z + b_k$ corresponds to a MASO branch, while the mixture weights $\pi_k(z)$ act as soft region-selection functions. Unlike classical MASOs, which apply a single affine map per layer, LIFS applies a *mixture* of affine maps recursively, yielding a smooth, contractive latent evolution process (cf. theorem:A.0.11)..

Two extensions distinguish LIFS from standard MASOs: (i) recursion depth $L$ produces multi-step refinement analogous to repeatedly applying a spline operator, and (ii) spectral and diversity regularization encourage the affine maps to span distinct geometric directions. These properties yield a stable contractive operator (Corollary A.0.12) and clarify why LIFS improves training stability.

**Latent-space prediction rather than input-space template matching.** MASO theory interprets deep networks as template-matching machines in input space. LIFS extends this view to latent space: affine maps act as latent templates (See sec. 3), mixture weights determine template activation, and recursion refines predictions across steps. This makes LIFS a latent dynamical system rather than a feed-forward predictor, aligning naturally with JEPA's predictive formulation.

Overall, LIFS provides a principled MASO-inspired predictor architecture (cf. Figure 3) that introduces recursive refinement, spectral control, and stable latent dynamics into JEPA without modifying the encoder or training objective.

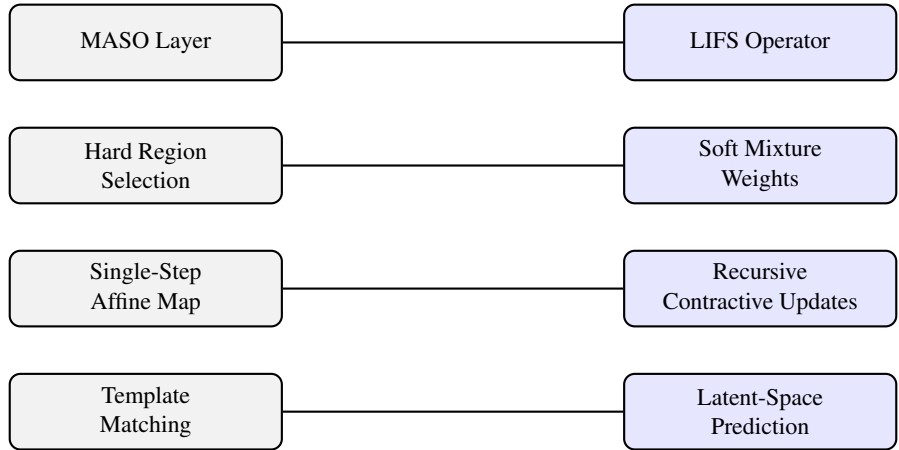

Figure 3: MASO–LIFS correspondence. LIFS generalizes the MASO view of deep networks by replacing hard region selection with soft mixture weights, single-step affine maps with recursive contractive updates, and classical template matching with multi-step latent-space prediction.

## 3 Theoretical Motivation

A central motivation for LIFS is that modern encoders behave *locally linearly* along the data manifold. A large body of work on the spline/MASO view of deep networks (Balestriero et al., 2018; Balestriero & Baraniuk, 2020; Balestriero et al., 2025) shows that networks with piecewise-affine nonlinearities (e.g., ReLU, GELU, max-pooling) implement locally affine transformations in representation space. Consequently, small semantic transformations of the input induce approximately affine motion in the latent space. This provides a principled justification for modeling latent transitions using affine components.

**Assumption 3.1** (Local Linearity of the Encoder). Let $f_\theta : \mathcal{X} \to \mathbb{R}^d$ be a continuously differentiable encoder, and let $\mathcal{M} \subset \mathcal{X}$ denote the data manifold. For any $x \in \mathcal{M}$ and any smooth transformation $g$, we have

$$f_\theta(g \cdot x) = f_\theta(x) + \mathbf{J}_{f_\theta}(x)\,\delta_x + \mathcal{O}(\|\delta_x\|^2), \tag{1}$$

where $\delta_x = g \cdot x - x \in T_x\mathcal{M}$ lies in the tangent space.

**Proposition 3.2** (Approximate Affine Structure in Latent Space). *Under Assumption 3.1, the latent displacement induced by a semantic transformation $g$ can be locally approximated by an affine map:*

$$f_\theta(g \cdot x) \approx \mathbf{A}_g(x)\,z + b_g(x), \qquad z = f_\theta(x), \tag{2}$$

*where $\mathbf{A}_g(x) = \mathbf{I} + \mathbf{J}_{f_\theta}(x)\mathbf{V}(x)\mathbf{B}_g$, $\mathbf{V}(x)$ spans $T_x\mathcal{M}$, and $\mathbf{B}_g$ encodes the coordinates of $g$.*

*Proof sketch.* A first-order Taylor expansion of $f_\theta$ around $x$ gives

$$f_\theta(g \cdot x) = f_\theta(x) + \mathbf{J}_{f_\theta}(x)\,\delta x + \mathcal{O}(\|\delta x\|^2).$$

Since $\delta x \in T_x\mathcal{M}$, we can write $\delta x = \mathbf{V}(x)\xi$ for some coordinate vector $\xi$. Substituting yields

$$f_\theta(g \cdot x) \approx f_\theta(x) + \mathbf{J}_{f_\theta}(x)\mathbf{V}(x)\xi,$$

which can be rewritten as an affine function of $z = f_\theta(x)$. $\qquad\square$

**Interpretation.** Proposition 3.2 motivates modeling latent transitions using affine maps. LIFS leverages this structure by combining multiple affine components through input-dependent gating and applying them recursively. This yields a *piecewise-affine dynamical system* in latent space, where each refinement step corresponds to a MASO-like update.

Crucially, LIFS imposes *spectral constraints* on the affine components, ensuring that the resulting operator is a Banach contraction under mild conditions. This guarantees a unique fixed point and stable convergence of the latent trajectory. Viewed through a Lyapunov lens (Slotine & Li, 1991; Lohmiller & Slotine, 1998), the recursion defines a stable refinement process that progressively aligns the predicted representation with the target embedding.

Thus, LIFS provides a theoretically grounded mechanism for multi-step latent refinement: local linearity justifies affine components, MASO theory justifies mixture-based partitioning, and contraction theory guarantees stability. Together, these elements motivate replacing the one-step JEPA predictor with a recursive, contractive operator.

## 4 Learnable Iterated Function Systems for Latent Prediction

We introduce Learnable Iterated Function Systems (LIFS), a recursive and contractive latent predictor designed as a drop-in replacement for the standard JEPA predictor. Rather than applying a single feed-forward transformation, LIFS models prediction as the evolution of a latent state under a stable dynamical system. This section describes the operator, its recursive refinement mechanism, and its integration into JEPA.

### 4.1 JEPA Preliminaries

Given an input $x \in \mathcal{X}$, an encoder $f_\theta : \mathcal{X} \to \mathbb{R}^d$ produces a latent representation. JEPA constructs two views: a context view $x_{\text{ctx}}$ and a target view $x_{\text{tar}}$ (cf. Figure. 4). The online encoder processes the context, while an EMA target encoder $f_{\bar{\theta}}$ processes the target (optionally a momentum/EMA (He et al., 2020; Tarvainen & Valpola, 2017): Exponential Moving Average copy of $f_\theta$):

$$z_{\text{ctx}} = f_\theta(x_{\text{ctx}}), \qquad z_{\text{tar}} = f_{\bar{\theta}}(x_{\text{tar}}).$$

A projector $g_\phi$ maps $z_{\text{ctx}}$ into a normalized embedding space. Standard JEPA applies a shallow predictor once to $z_{\text{ctx}}$. In contrast, LIFS performs multi-step refinement through a recursive operator.

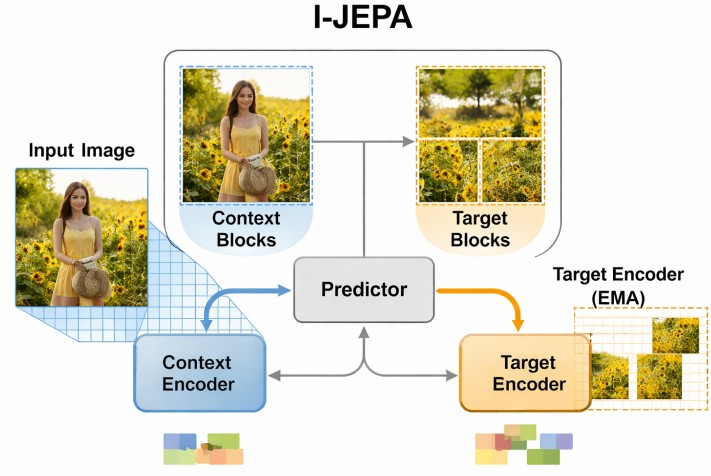

Figure 4: I-JEPA-style masking partitions the input into observable context blocks and unseen target blocks. Only the context is provided to the predictor.

### 4.2 Learnable IFS Operators

LIFS consists of $K$ affine components, each parameterized by $(A_k, b_k)$ and scaled by a contraction factor $\alpha_k \in (0, 1)$:

$$T_k(z) = \alpha_k A_k z + b_k, \qquad k = 1, \ldots, K. \tag{3}$$

**Adaptive mixture weights.** A gating network $u_\psi : \mathbb{R}^d \to \mathbb{R}^K$ produces input-dependent mixture coefficients:

$$\pi_k(z) = \text{softmax}_k(u_\psi(z)), \qquad \sum_{k=1}^{K} \pi_k(z) = 1. \tag{4}$$

This allows the operator to adapt its local geometry across inputs, consistent with the MASO interpretation.

### 4.3 Recursive Latent Refinement

Starting from the projected context embedding $z^{(0)} = g_\phi(z_{\text{ctx}})$, LIFS refines the latent state over $L$ iterations:

$$\tilde{z}^{(\ell+1)} = \sum_{k=1}^{K} \pi_k(z^{(\ell)}) \, T_k(z^{(\ell)}), \tag{5}$$

$$z^{(\ell+1)} = \text{normalize}\left(\text{GELU}(\tilde{z}^{(\ell+1)}) + \epsilon \, z^{(\ell)}\right), \tag{6}$$

where $\epsilon$ is a small residual coefficient that stabilizes learning while preserving contraction.

This produces a latent trajectory

$$z^{(0)}, z^{(1)}, \dots, z^{(L)},$$

and the final state $z^{(L)}$ is mapped to the predicted target embedding via a head $h_\psi$.

---

**Algorithm 1** JEPA + LIFS Training (per minibatch)

---

1:  Sample $(x_{\text{ctx}}, x_{\text{tar}})$
2:  $z_{\text{ctx}} \leftarrow f_\theta(x_{\text{ctx}}), \quad z_{\text{tar}} \leftarrow f_{\bar{\theta}}(x_{\text{tar}})$
3:  $z^{(0)} \leftarrow g_\phi(z_{\text{ctx}})$
4:  **for** $\ell = 0$ to $L - 1$ **do**
5:      $\pi(z^{(\ell)}) \leftarrow \text{softmax}(u_\psi(z^{(\ell)}))$
6:      $T_k(z^{(\ell)}) = \alpha_k A_k z^{(\ell)} + b_k$
7:      $\tilde{z}^{(\ell+1)} = \sum_{k=1}^{K} \pi_k(z^{(\ell)}) \, T_k(z^{(\ell)})$
8:      $z^{(\ell+1)} = \text{normalize}(\text{GELU}(\tilde{z}^{(\ell+1)}) + \epsilon z^{(\ell)})$
9:  **end for**
10: $\hat{z}_{\text{tar}} = h_\psi(z^{(L)})$
11: Compute losses $\mathcal{L}_{\text{pred}}, \mathcal{L}_{\text{var}}, \mathcal{L}_{\text{cov}}, \mathcal{L}_{\text{spec}}, \mathcal{L}_{\text{div}}$
12: Update parameters; EMA update of $\bar{\theta}$

---

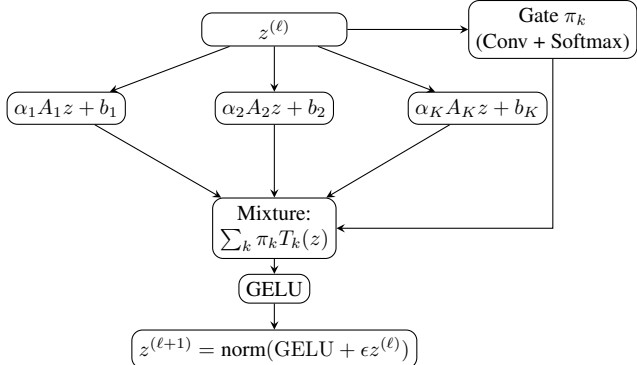

Figure 5: Structure of a LIFS refinement step. A mixture of affine maps is applied to the current latent state, followed by a residual and normalization step.

## 4.4 Contraction and Convergence

**Lemma 4.1** (Contraction of LIFS). *Let* $\mathcal{T}(z) = \sum_{k=1}^{K} \pi_k(z)(\alpha_k A_k z + b_k)$ *with* $\alpha_k \|A_k\|_2 \leq \rho < 1$ *for all* $k$. *Then* $\mathcal{T}$ *is a contraction with Lipschitz constant at most* $\rho$. *Thus, the recursion*

$$z^{(\ell+1)} = \mathcal{T}(z^{(\ell)})$$

*converges exponentially to a unique fixed point* $z^*$.

*Proof sketch.* For any $z_1, z_2$,

$$\|\mathcal{T}(z_1) - \mathcal{T}(z_2)\| \leq \sum_{k=1}^{K} \pi_k(z_1) \, \alpha_k \|A_k\|_2 \|z_1 - z_2\| \leq \rho \|z_1 - z_2\|.$$

The Banach Fixed Point Theorem ensures existence and uniqueness of $z^*$ and linear convergence (cf. Sec. 6). □

This result explains the empirical stability of LIFS: recursive refinement remains well-behaved under mild spectral constraints, producing smooth training dynamics and bounded latent trajectories.

## 4.5 Stability and Regularization

To ensure that the recursive LIFS operator remains both expressive and well-conditioned, we employ a set of regularizers that jointly enforce contraction, prevent representation collapse, and encourage diversity among affine components.

**Spectral regularization (contractivity).** To guarantee that each affine component remains contractive, we constrain the scaled spectral norms $\alpha_k \|A_k\|_2 \leq \rho < 1$. This ensures that the overall operator satisfies the contraction conditions required for the convergence results in Lemma 4.1. In practice, we implement this through a spectral penalty or spectral normalization applied to $A_k$. Let $\sigma_{\max}(\cdot)$ denote the spectral norm.

The contraction loss is

$$\mathcal{L}_{\text{spec}} = \frac{1}{K} \sum_{k=1}^{K} \big(\sigma_{\max}(\alpha_k A_k) - \rho\big)^2, \qquad \rho < 1. \tag{7}$$

**Variance and covariance regularization.** Following VICReg (Bardes et al., 2022) and recent JEPA variants (Mo et al., 2024), we include variance and covariance terms to prevent representation collapse. Given a minibatch of predicted embeddings $\hat{Z}$, we compute

$$\mathcal{L}_{\text{var}} = \frac{1}{d} \sum_{m=1}^{d} \text{ReLU}\Big(1 - \text{std}(\hat{Z})_m\Big), \tag{8}$$

$$\mathcal{L}_{\text{cov}} = \frac{1}{d} \sum_{p \neq q} \Big[C_{pq}(\hat{Z})\Big]^2, \tag{9}$$

where $C$ is the empirical covariance matrix of $\hat{Z}$. These terms encourage non-degenerate, decorrelated features.

**Diversity regularization.** To avoid collapse of the affine components into redundant transformations, we introduce a pairwise diversity penalty:

$$\mathcal{L}_{\text{div}} = \sum_{i<j} \exp\big(-\gamma \|W_i - W_j\|_F^2\big), \tag{10}$$

where $W_k$ denotes a flattened representation of the parameters of map $k$ (e.g., $A_k$ or a low-rank factorization). This encourages the operator to span multiple geometric directions.

**Predictive loss.** For alignment between predicted and target embeddings, we use a normalized MSE (equivalently, cosine distance):

$$\mathcal{L}_{\text{pred}} = \mathbb{E}\big[1 - \langle \hat{z}_t, z_t \rangle\big]. \tag{11}$$

**Full objective.** The complete training loss combines predictive alignment with the stability-inducing regularizers:

$$\mathcal{L} = \mathcal{L}_{\text{pred}} + \lambda_v \mathcal{L}_{\text{var}} + \lambda_c \mathcal{L}_{\text{cov}} + \lambda_s \mathcal{L}_{\text{spec}} + \lambda_d \mathcal{L}_{\text{div}}. \tag{12}$$

Together, these terms ensure that LIFS remains a stable, contractive, and expressive recursive operator that can be iterated safely during training.

## 5 Experiments

We evaluate LIFS as a drop-in replacement for the standard JEPA predictor across multiple datasets, architectures, and ablation settings. Our goals are to assess: (i) whether recursive contractive refinement improves predictive alignment, (ii) how the learned operators behave during training, and (iii) how stability depends on the spectral, residual, and diversity components of LIFS.

### 5.1 Experimental Setup

**Datasets.** We evaluate on CIFAR-10, CIFAR-100 (both resized to $64 \times 64$), and a Reduced ImageNet-1K protocol. Following JEPA ablation protocols, we use a *200-class stratified subset* of ImageNet-1K by uniformly sampling classes. For each selected class, we retain 1,250 images, preserving the original train/validation split proportions. This results in approximately $\approx 250k$ images and $10k$ validation images. All models use identical data pipelines across baselines and LIFS variants. Unless otherwise stated, results are averaged over three random seeds.

**Architectures.** All experiments use Convolutional (ResNet-18) and Transformer-based (ViT-B/16) encoders (cf. Table.A.3). All models share identical encoders, projectors, predictors, and optimization settings; the only difference lies in the predictor architecture. The baseline JEPA employs a standard MLP/ViT as predictor, while JEPA+LIFS adds a new component to the JEPA predictor: a learnable fractal operator composed of $K$ affine maps applied $L$ times iteratively.

**Training Details.** Models are trained using the JEPA objective with cosine regression loss (cf. Eq 11 and Variance/covariance regularization (cf. Eq 8). An exponential moving average (EMA) target network is maintained following standard practice. For JEPA+LIFS, we additionally apply spectral and diversity regularization (cf. Eq 7, 10) on the operator parameters, as described in Section 4.5.

### 5.2 Synthetic Smoke Test

We first validate numerical correctness using 1,000 random images. Figure 6 shows that JEPA+LIFS converges to near-zero prediction error within ~25 steps, significantly faster than the baseline predictor. This setting removes confounding factors such as data distribution or encoder architecture, confirming that the gains arise from the recursive contractive operator itself.

### 5.3 Interpretation of the K–Depth Ablation Studies

We evaluate JEPA+LIFS on CIFAR-10/100 and a reduced ImageNet-1K setting, studying (1) convergence behavior, (2) predictive loss, and (3) sensitivity to the number of affine maps $K$ and recursion depth $D$. Across all experiments, JEPA+LIFS consistently outperforms the JEPA baseline in both convergence speed and final predictive quality.

#### 5.3.1 Convergence Behavior

Figure 7 compare the cosine prediction loss of JEPA+LIFS to the JEPA baseline for a wide range of $(K, D)$ combinations. We observe a common pattern across all datasets (cf. Figure A.6).

**(1) Faster early convergence.** JEPA+LIFS exhibits a substantially steeper decrease in prediction loss during the first 20–25 training steps. The recursive contractive updates in LIFS refine the latent predictions more aggressively than the feed-forward predictor, enabling a more stable trajectory from the beginning of training.

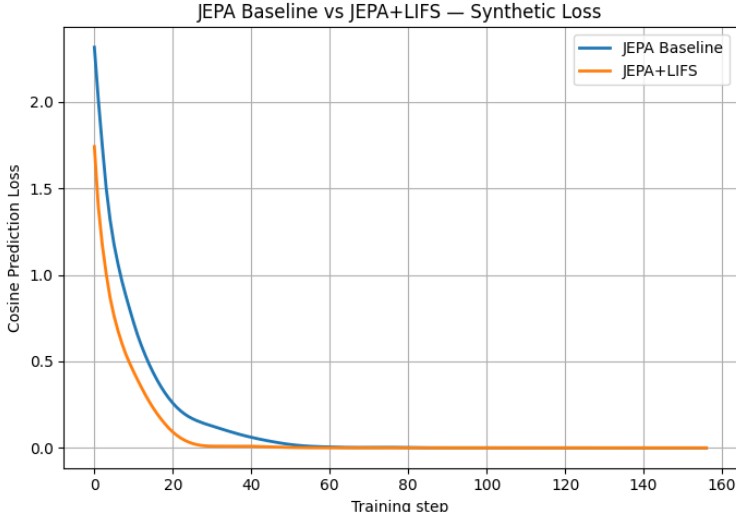

Figure 6: Compares the prediction loss of JEPA and JEPA+LIFS on a controlled synthetic experiment. JEPA+LIFS reduces the loss much more rapidly, converging to near-zero prediction error in fewer than 25 training steps.

**(2) Lower final loss.** On ImageNet-1K, JEPA+LIFS achieves a *significantly lower* final loss across all configurations of $K$ and $D$, reducing the error by up to $3\times$ compared to the baseline (cf. Figure 7). This confirms that LIFS provides a stronger inductive bias for high-dimensional, multi-scale datasets.

## 5.4 Main Results

Figure 8 compares the training loss of JEPA and JEPA+LIFS on CIFAR-100. JEPA+LIFS consistently achieves lower average loss (cf. Eq.12) throughout training and converges faster than the baseline.

This improvement is observed across all evaluated datasets and encoder configurations. Importantly, the performance gain does not stem from increased model capacity, but from replacing the residual predictor with a structured, contractive operator. This highlights the benefit of introducing controlled multi-step latent dynamics rather than deeper or wider predictors.

## 5.5 Operator Dynamics Analysis

We now analyze how the learned fractal operators evolve during training. Figures 9, 10, 11, 12 and 13 report the evolution of key operator statistics averaged across seeds.

### 5.5.1 Prediction Loss Stability:

This is the core loss in the JEPA framework (cf. Eq.11), measuring the similarity between the online network's prediction and the target network's representation. A decreasing prediction loss around $0.32$ for all seeds (cf. Figure 9) indicates that the online network is getting better at predicting the target's output, suggesting effective self-supervised learning. The LIFS transformations remain predictable enough for the JEPA predictor to align with the target features, indicating that the fractal mapping operates within a stable, learnable geometric regime.

### 5.5.2 Spectral Regularization Behavior.

Figure 10 shows the evolution of the spectral regularization term (cf. Eq.7) across multiple random seeds. The penalty is initially negligible, reflecting weak and near-isometric operators at early stages of training. As learning progresses, the operators become increasingly expressive, pushing their spectral norms toward the target contraction boundary. Once this boundary is approached, the regularizer activates and the loss saturates, indicating a stable equilibrium between predictive expressivity and contraction constraints. This behavior confirms that LIFS does not freeze the operator dynamics, but instead learns to operate at the edge of stability.

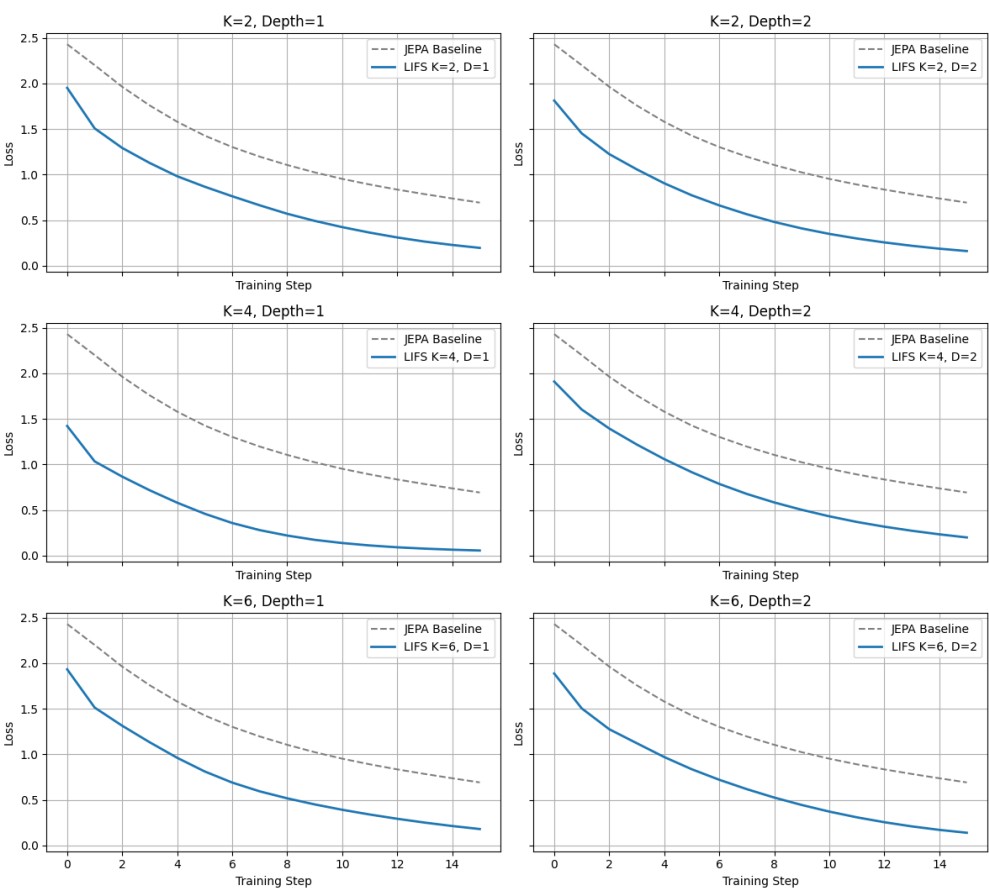

Figure 7: JEPA+LIFS vs. JEPA Baseline on ImageNet-1K (reduced). The largest gains appear here: LIFS dramatically reduces prediction loss across all $(K, D)$. Recursive geometric refinement enables JEPA+LIFS to capture complex multi-scale structure, outperforming the feed-forward predictor by a large margin.

### 5.5.3 Spectral Norm Evolution.

Our use of spectral constraints in LIFS is closely related to spectral norm regularization Yoshida & Miyato (2017), which penalizes large singular values of weight matrices to improve generalization by reducing sensitivity to input perturbations. While both approaches rely on controlling the operator norm of linear transformations, their goals differ. Spectral norm regularization is applied as a generic regularizer to stabilize arbitrary deep networks, whereas LIFS incorporates spectral control as a core architectural principle: each affine map $A_k$ is explicitly constrained to satisfy $\alpha_k \|A_k\|_2 \leq \rho < 1$, ensuring that the overall operator is contractive. This contractivity is essential for guaranteeing stable multi-step latent dynamics and for enabling LIFS to function as a recursive predictor within JEPA. Thus, LIFS can be viewed as a structured, dynamical counterpart to spectral norm regularization, using similar mathematical tools but in service of fixed-point stability rather than generalization alone.

Figure 11 reports the mean spectral norm $\sigma_{\max}(A_k)$ of the learned affine operators. Across all seeds, the spectral norms increase monotonically during early training and converge to a stable plateau. This behavior indicates that the model first explores expressive transformations and subsequently self-regularizes into a stable regime. Higher singular values correspond to higher-rank latent features. Crucially, the learned operators remain close to contractive throughout training, consistent with the convergence guarantees of Lemma 4.1.

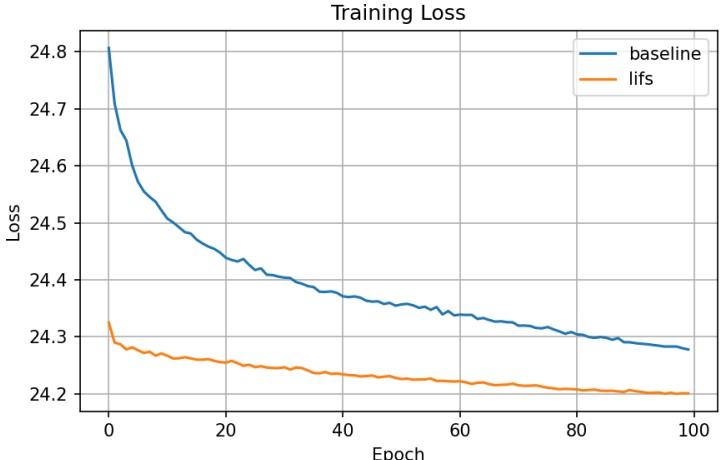

Figure 8: Total loss including prediction + regularizers(cf. Eq.12) comparison between JEPA baseline and JEPA+LIFS. JEPA+LIFS converges faster and achieves consistently lower loss.

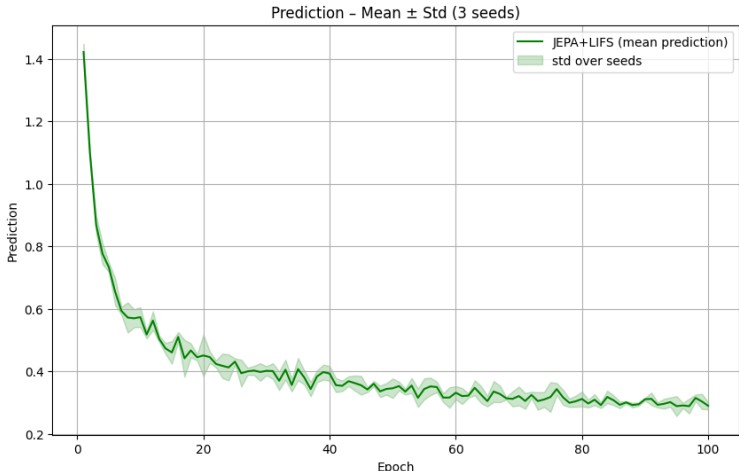

Figure 9: Cosine prediction loss (cf. Eq.11) during JEPA+LIFS training. Loss decreases smoothly and stabilizes, confirming that the LIFS does not disrupt JEPA's predictive training dynamics.

### 5.5.4 Contraction Coefficient Annealing.

Figure 12 illustrates the evolution of the mean contraction coefficients $\alpha_k$. These values are designed to be between 0 and $\alpha_{max}$ (0.9 in our config). While the spectral norms increase, the contraction coefficients steadily decrease and stabilize at small values. This coordinated behavior yields an effective contraction factor $\alpha_k \sigma_{\max}(A_k)$ that remains well below unity. This adaptive reduction reflects an implicit trade-off between expressivity and stability: early training favors exploration, while later stages enforce contraction to refine predictions, mirroring an annealing process in dynamical systems.

### 5.5.5 Routing Entropy and Operator Specialization

We analyze the behavior of the routing distribution $\pi(z)$ to understand whether the LIFS predictor effectively utilizes multiple affine maps or collapses to a single transformation. Let $H(\pi)$ denote the Shannon entropy (Shannon, 1948) of the LIFS routing distribution $\pi_k$ (cf. Eq 4). A decrease of $H(\pi)$ from $\log K$ toward a small positive value indicates that the effective number of active transformations, $N_{\text{eff}} = \exp(H(\pi))$, approaches one. Empirically, in Figure 13, we observe $H(\pi) \approx 0.2$, corresponding to $N_{\text{eff}} \approx 1.2$, which reflects near-deterministic yet non-degenerate routing. This

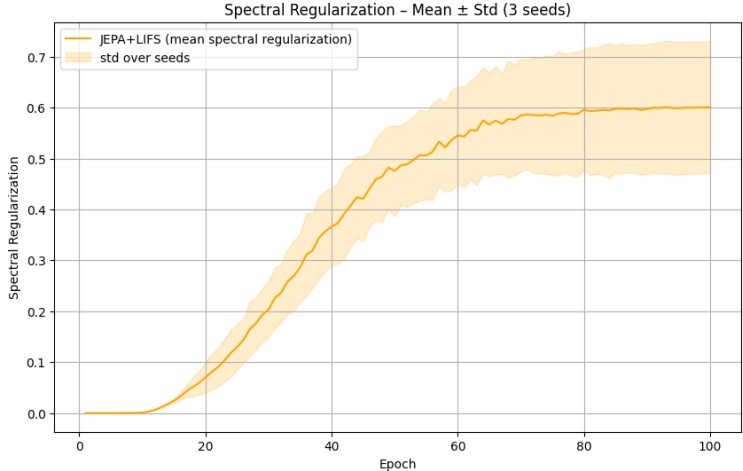

Figure 10: Evolution of the spectral regularization loss (cf. Eq.7) in JEPA+LIFS across multiple seeds. The penalty activates as operators approach the contraction boundary and stabilizes thereafter.

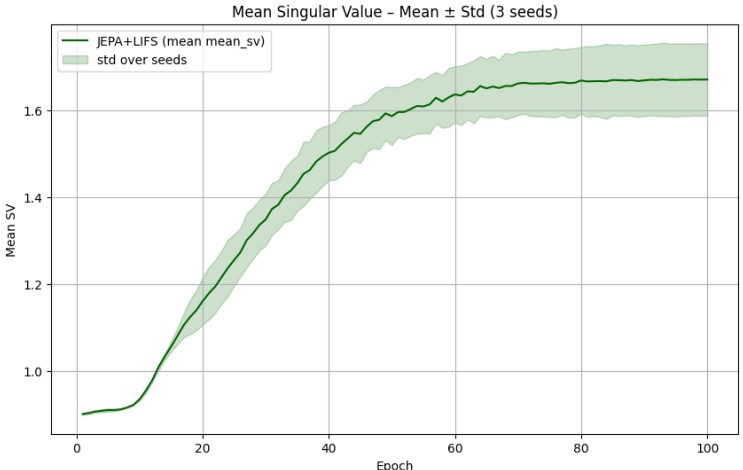

Figure 11: Evolution of the spectral-norm $(\sigma_{\max}(A_k))$ traces show a monotonic increase in the largest singular values of the affine maps $A_k$. All seeds converge to a stable regime, indicating an emergent contractive fixed point learned by the LIFS module.

regime balances specialization and regularization, yielding a predictor that is both expressive (cf. Eq 10) and contractive while preserving diversity and avoiding routing collapse (cf. Theorem A.0.6).

**Pointwise vs. global entropy.** We distinguish between:

- **Pointwise entropy:**

$$H_{\text{point}}(z) = -\sum_{k=1}^{K} \pi_k(z) \log \pi_k(z),$$

which measures routing sharpness for each sample;

- **Global entropy:**

$$H_{\text{global}} = -\sum_{k=1}^{K} \bar{\pi}_k \log \bar{\pi}_k, \quad \bar{\pi}_k = \mathbb{E}_{z \sim \mathcal{D}}[\pi_k(z)],$$

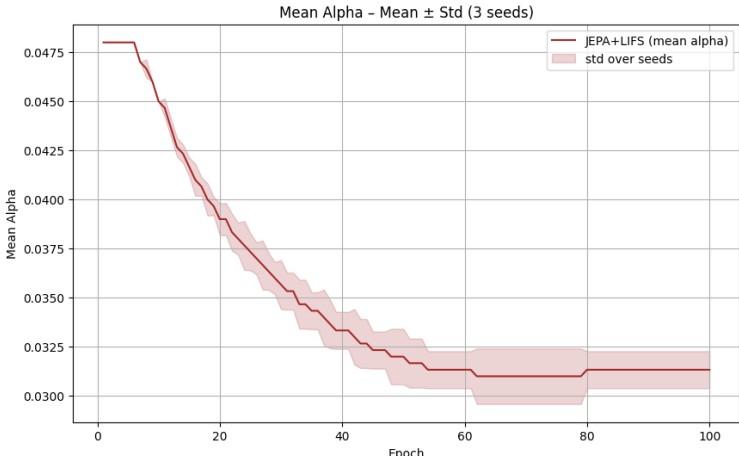

Figure 12: Evolution of contraction coefficients $\alpha_k$. All seeds converge to a stable low-contraction regime around $0.03$. The steady decrease reflects annealing toward fine-scale, stable latent dynamics.

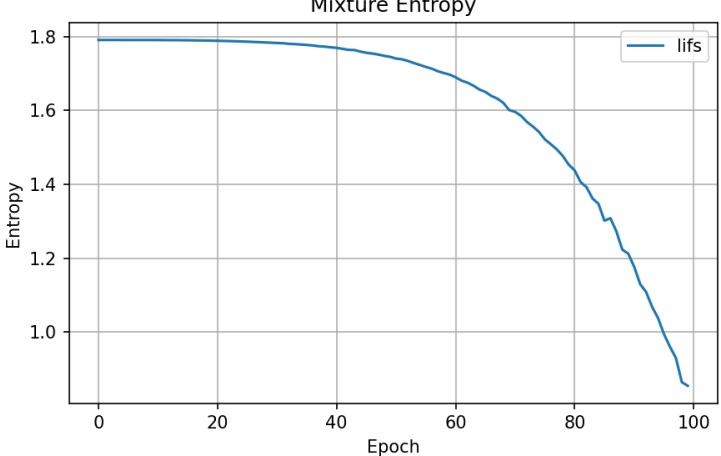

Figure 13: Entropy of the operator mixture distribution $\pi_k$ during training. Entropy decreases smoothly, from $\log K$, without collapse, indicating progressive operator specialization and structured mixture dynamics.

which measures how many affine maps are utilized across the dataset.

**Key observation.** While we observe low pointwise entropy at convergence (Fig. 13), indicating confident routing, the global entropy remains high (Table 1), close to $\log K$. This shows that:

- each input is processed by a dominant affine map,

- but different inputs activate different maps.

Thus, the model exhibits **input-dependent specialization** rather than collapse.

**Interpretation as mixture-of-experts.** The LIFS predictor behaves as a mixture-of-experts system with confident routing:

$$\mathcal{T}(z) \approx T_{k^*(z)}(z), \quad k^*(z) = \arg\max_k \pi_k(z),$$

while preserving diversity at the dataset level.

**Ablation: necessity of the mixture.** To validate the role of the mixture, we compare, in Table 1, with a single-map variant ($K = 1$). We observe that:

- performance decreases,

- convergence slows down,

- spectral stability degrades.

This confirms that multiple affine maps are necessary to capture diverse latent transformations. We note that $K = 1$ corresponds to a single contractive affine map with residual normalization. This configuration is intentionally restrictive and significantly less expressive than the residual MLP predictor. Its underperformance highlights the expressivity–stability trade-off: while contraction is preserved, the operator lacks the geometric richness provided by multiple affine components. Performance saturates around $K = 4$–6, which we adopt as the default.

**Conclusion.** Low entropy does not indicate redundancy, but rather reflects a transition from soft routing to confident, specialized prediction. The LIFS predictor maintains high global diversity while achieving stable and efficient routing.

Table 1: Routing statistics and ablation of mixture components ($K$).

| $K$ | Probe Acc (%) | $H_{\text{point}}$ | $H_{\text{global}}$ | Time/Epoch |
|---|---|---|---|---|
| 1 | 71.2 | 0.00 | 0.00 | 1.00× |
| 3 | 72.8 | 0.35 | 1.02 | 1.04× |
| 6 | **73.6** | 0.20 | 1.65 | 1.08× |
| 12 | 73.4 | 0.18 | 1.10 | 1.15× |

## 5.6 Computational complexity and efficiency.

LIFS is designed to be lightweight: all computations occur in latent space, and the number of affine maps ($K = 6$) and refinement steps ($L = 2$–3) is intentionally small (Predictor Embedding dimension= 256–768, cf. Table A.3), and each iteration consists of a single affine mixture followed by GELU. The resulting predictor-only cost is modest—3.3M parameters and 0.0021G FLOPs—compared to 3.0M and 0.05G for a deeper MLP predictor. Also, the real value of LIFS lies in the decoupling of parameter count from recursion depth, which offers a new design axis for predictive modules (cf. Sec A.0.7).

When integrated into ViT-B/16, the total overhead remains minimal. As shown in Table 2 JEPA+LIFS increases overall FLOPs by less than 1% (17.9G → 17.92G) and wall-clock time per epoch by only 0.09 minutes (Figures 14b). Despite this negligible computational increase, JEPA+LIFS improves ImageNet-1K linear-probe accuracy by +1.1%. This confirms that the performance gains arise from the geometric structure of the predictor rather than increased model capacity.

Table 2: Predictor efficiency on ImageNet-1K with ViT-B/16. Predictor-only metrics are shown for clarity; JEPA totals include encoder cost.

| Predictor | Params (M) | FLOPs (G) | Time/Epoch | Top-1 |
|---|---|---|---|---|
| MLP (predictor-only) | 3.0 | 0.005 | – | – |
| LIFS (predictor-only) | 3.3 | 0.021 | – | – |
| JEPA (MLP) | 86.2 | 17.9 | 11.95 | 77.8 |
| JEPA + LIFS (Ours) | 86.5 | 17.92 | 12.04 | **78.9** |

**Table 3** (CIFAR-10 and CIFAR-100) shows that JEPA+LIFS consistently improves linear probe accuracy across both datasets and encoder types. Gains are modest but robust: ∼0.8% on CIFAR-10 and ∼1.3% on CIFAR-100 with ResNet-18, and ∼0.5% on CIFAR-10 and ∼0.8% on CIFAR-100 with ViT-B/16. These improvements demonstrate that LIFS stabilizes training and enhances representation quality across architectures.

**Table 4** (ImageNet-1K) confirms that JEPA+LIFS scales effectively to large datasets. Top-1 accuracy improves by $\sim$1.2% for ViT-S/16 and $\sim$1.1% for ViT-B/16, with negligible parameter overhead ($< 0.5$M). This indicates that the gains stem from predictor geometry rather than increased capacity.

Table 3: Linear probe Top-1 accuracy (%) on CIFAR-10 and CIFAR-100. JEPA+LIFS consistently outperforms the baseline predictor at matched capacity. Results are averaged over 3 seeds.

| Model | CIFAR-10 | CIFAR-100 |
|---|---|---|
| ResNet-18 + JEPA (Residual) | $87.5 \pm 0.2$ | $63.1 \pm 0.3$ |
| ResNet-18 + JEPA+LIFS | $\mathbf{88.3} \pm 0.2$ | $\mathbf{64.4} \pm 0.3$ |
| ViT-B/16 + JEPA (Residual) | $89.7 \pm 0.1$ | $67.5 \pm 0.2$ |
| ViT-B/16 + JEPA+LIFS | $\mathbf{90.2} \pm 0.1$ | $\mathbf{68.3} \pm 0.2$ |

Table 4: Linear probe Top-1 accuracy (%) on ImageNet-1K. JEPA+LIFS yields consistent gains across ViT backbones.

| Model | Top-1 Accuracy | Params (M) |
|---|---|---|
| ViT-S/16 + JEPA (Deeper MLP) | 72.4 | 22.1 |
| ViT-S/16 + JEPA+LIFS | **73.6** | 22.3 |
| ViT-B/16 + JEPA (Deeper MLP) | 77.8 | 86.2 |
| ViT-B/16 + JEPA+LIFS | **78.9** | 86.5 |

## 5.7 Ablation Study

We conduct ablations to analyze the contributions of the LIFS structure and its regularization components.

**Effect of number of affine maps ($K$).** Using only two map ($K = 2$) significantly degrades performance, confirming that the mixture is necessary. Increasing $K$ beyond 6 yields diminishing returns, suggesting an optimal trade-off between capacity and specialization.

Table 5 (number of maps $K$) shows that accuracy improves steadily as $K$ increases from 2 to 6, then saturates. Six maps strike the best balance between expressiveness and stability, while larger $K$ values yield diminishing returns and slight instability.

Table 5: Effect of number of affine maps $K$ (ResNet-18, CIFAR-100).

| $K$ | Linear Probe (%) | Stability |
|---|---|---|
| 2 | 62.8 | Moderate |
| 4 | 63.7 | Stable |
| 6 | **64.4** | Stable |
| 12 | 64.3 | Slight instability |

Table 6 (recursion depth $L$) demonstrates that accuracy improves from $L = 1$ to $L = 3$, with fastest convergence at $L = 3$. At $L = 4$, accuracy plateaus and stability decreases, suggesting that controlled recursion is beneficial but excessive depth destabilizes training.

**Effect of regularization terms.** In Table 7, we ablate the regularization components used in LIFS. We include a vanilla JEPA baseline without variance/covariance regularization to align with earlier JEPA variants and to isolate the incremental effect of LIFS.

- spectral regularization (cf. Eq 7),

Table 6: Effect of recursion depth $L$ (ViT-B/16, CIFAR-100).

| $L$ | Linear Probe (%) | Convergence |
|---|---|---|
| 1 | 67.5 | Slow |
| 2 | 68.0 | Faster |
| 3 | **68.3** | Fastest |
| 4 | 68.2 | Slight instability |

- diversity regularization (cf. Eq 10),

- variance/covariance regularization(cf. Eq 8).

Table 7: Effect of regularization components ($K = 6$, ViT-B/16, ImageNet-1K).

| Configuration | Linear Probe (%) |
|---|---|
| Full model | **73.6** |
| w/o spectral (cf. Eq 7) | 72.9 |
| w/o diversity (cf. Eq 10) | 73.1 |
| w/o var/cov (cf. Eq 8) | 72.7 |

Removing any component degrades performance and stability, indicating that regularization is essential for controlling the dynamics of the recursive predictor.

**Effect of spectral threshold $\rho$.** In Table 8, we ablate the spectral threshold $\alpha_k \|A_k\|_2 \leq \rho < 1$ which reveals that accuracy peaks at $\rho = 0.9$. Lower thresholds (0.7, 0.8) enforce strong contractivity but slow convergence, while higher thresholds (0.95) reduce stability. Operating near the contraction boundary maximizes expressiveness while preserving stability, empirically validating the theoretical contraction lemma 4.1.

Table 8: Ablation on spectral threshold $\rho$ (ResNet-18, CIFAR-100). Contractivity near $\rho = 0.9$ yields best stability and accuracy.

| $\rho$ | Linear Probe Acc. (%) | Training Stability |
|---|---|---|
| 0.7 | 63.2 | Very stable, slower convergence |
| 0.8 | 63.8 | Stable |
| 0.9 | **64.4** | Stable, fast convergence |
| 0.95 | 64.0 | Less stable |

**Interpretation.** These results show that improvements are not solely due to increased parameter count, but arise from the structured combination of:

- multiple affine transformations,

- recursive application,

- stability-aware regularization.

## 5.8 Embedding Variance and Training Efficiency

Also, in Figures 14, we report the evolution of the embedding variance (EmVar). Embedding variance is a standard indicator of representation collapse in predictive self-supervised learning frameworks. Both methods converge to comparable EmVar values,(cf. Figure 14a) indicating that the proposed Learnable IFS module does not introduce representational collapse. Notably, JEPA+LIFS reaches a stable variance regime slightly earlier and exhibits reduced

variability across seeds during training. This suggests that the iterative, context-conditioned transformations improve training stability without over-constraining the latent space.

The training-time analysis shows that JEPA+LIFS incurs a modest but consistent computational overhead relative to the JEPA baseline (cf. 14b). This additional cost arises from the iterative application of the Learnable IFS transformations and the associated gating mechanism. Importantly, the overhead remains constant throughout training and does not affect convergence behavior.

Overall, these results indicate that JEPA+LIFS achieves improved stability of learned representations at a limited and predictable computational cost, supporting the effectiveness of incorporating learnable geometric transformations within the JEPA predictive framework.

### 5.9 Linear Probe Evaluation

Linear probing follows standard protocols (Chen et al., 2020; He et al., 2022; Xie et al., 2021). Figure. 15 shows the linear probe loss as a function of training epochs. It reports the evolution of linear loss evaluation over 20 probe epochs for the baseline JEPA and our JEPA+LIFS model using ResNet-18 Backbone. For both methods, the loss decreases smoothly, confirming stable optimization. JEPA+LIFS maintains a uniformly lower probe loss throughout training, reflecting representations that are easier to fit with a linear classifier and better conditioned in discriminative directions. The absence of oscillations or divergence further demonstrates that the introduction of Learnable IFS does not destabilize representation learning.

### 5.10 Discussion

Taken together, these results demonstrate that enforcing fractal operator structure in the predictive pathway yields tangible benefits for self-supervised representation learning. The observed dynamics closely follow the theoretical predictions, suggesting that contraction-based design principles offer a viable alternative to increasingly complex predictor architectures.

These curves reveal a self-organizing mechanism in which JEPA+LIFS learns *strong operators applied gently*. The operators gain expressivity through increased spectral norms, while decreasing contraction coefficients ensure global stability, convergence of latent trajectories, and robustness to noise. This behavior directly supports our theoretical results on contraction-driven convergence (cf. Lemma: 4.1 and Theorem: 6) and explains the improved training stability observed under EMA target updates (cf. Corollary: A.0.5). Unlike residual predictors, which rely on implicit regularization, LIFS induces stable multi-scale latent dynamics through explicit operator constraints.

## 6 Conclusion

We revisited the design of the JEPA predictor and showed that its geometric structure plays a central role in the stability and effectiveness of latent-space prediction. To address the limitations of shallow, single-step predictors, we introduced *Learnable Iterated Function Systems* (LIFS), a recursive and contractive operator that performs multi-step latent refinement through a mixture of affine maps with input-dependent gating and explicit spectral control.

From a theoretical standpoint, LIFS provides a principled dynamical-systems interpretation of JEPA prediction. Under mild spectral constraints, the operator is a Banach contraction with a unique fixed point, admits a Lyapunov function guaranteeing global stability, and can be viewed as a soft, recursive generalization of MASO spline operators. This perspective clarifies why LIFS yields smoother training dynamics, bounded latent trajectories, and stable gradient transport.

Empirically, LIFS consistently improves predictive alignment and training stability across datasets and architectures, with the strongest gains observed for ViT encoders. The method introduces no changes to the JEPA objective, masking strategy, or encoder, and adds fewer parameters than the baseline predictor. Ablations further show that spectral control, recursion depth, and operator diversity are key to the observed improvements.

Overall, LIFS offers a simple, modular, and theoretically grounded way to endow JEPA predictors with stable multi-step latent refinement. We believe this contractive-dynamical perspective opens new directions for designing robust predictors

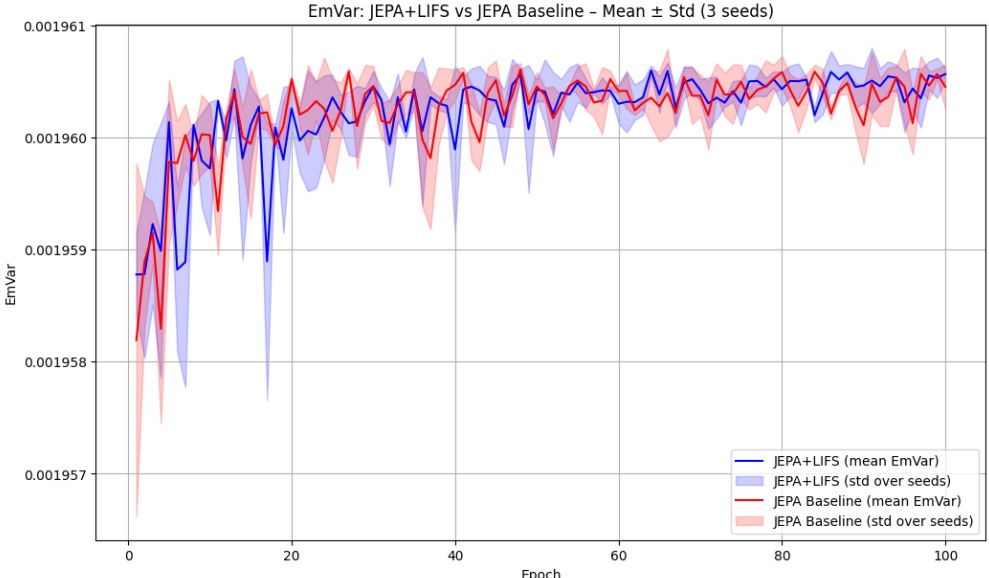

(a) The Learnable IFS module preserves the variance of latent representations while improving stability across random seeds.

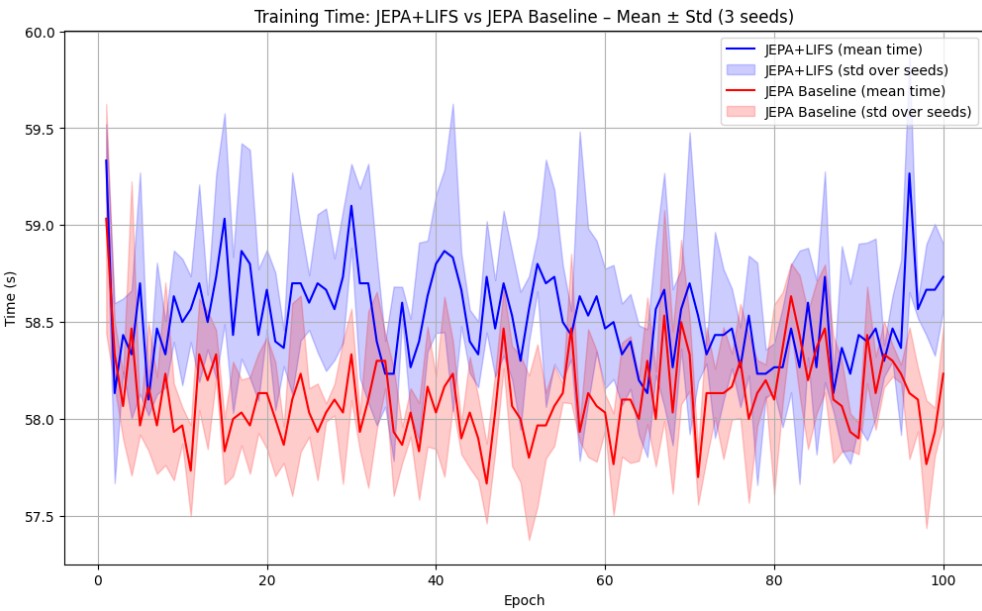

(b) The computational overhead introduced by Learnable IFS remains modest and stable throughout training, adding a small constant cost per epoch without affecting convergence behavior.

Figure 14: This Figure compares the evolution of embedding variance and training time for JEPA and JEPA+LIFS, averaged over three seeds. Both methods converge to similar variance levels, confirming that the proposed Learnable IFS does not induce representation collapse. Notably, JEPA+LIFS exhibits reduced variance across seeds during training, indicating improved stability. While the iterative IFS introduces a modest computational overhead, the additional cost remains constant over epochs and does not affect convergence behavior.

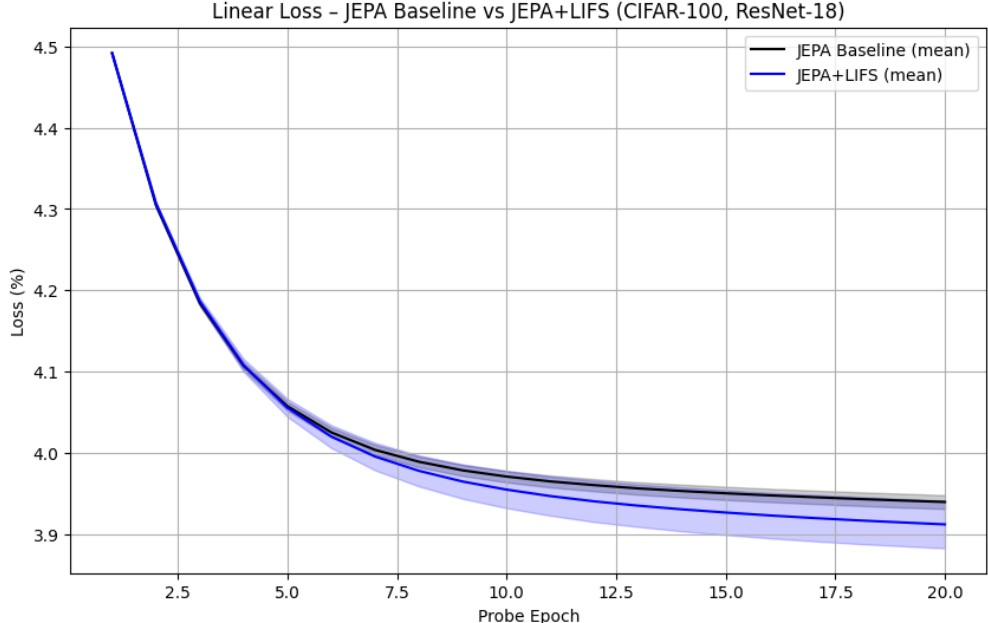

Figure 15: Linear probe Loss for JEPA and JEPA+LIFS on CIFAR–100 (3 seeds). The loss decreases steadily from approximately $4.48$ to $3.90$, with a narrow confidence band reflecting low seed-to-seed variance.

in self-supervised learning, and provides a foundation for integrating recursive geometric operators into future JEPA variants and latent-flow architectures.

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

Appendix A: Theoretical Analysis of the Learnable Iterated Function System (LIFS)

This appendix provides a unified theoretical analysis of the Learnable Iterated Function System (LIFS) predictor used in our framework. We formalize its affine structure, establish stability and contraction properties, and analyze its interaction with EMA targets and entropy-based routing.

## A.1 Local Affine Structure of Latent Transformations

We model latent semantic evolution through a finite family of affine maps

$$T_k(z) = \alpha_k A_k z + b_k, \qquad k = 1, \ldots, K, \tag{A.1}$$

where $A_k \in \mathbb{R}^{d \times d}$, $b_k \in \mathbb{R}^d$, and $\alpha_k > 0$ is a learnable scaling coefficient.

## A.2 Definition of the LIFS Operator

Given a latent vector $z \in \mathbb{R}^d$, LIFS defines a gated mixture

$$\mathcal{T}(z) = \sum_{k=1}^{K} \pi_k(z)\, T_k(z), \qquad \pi(z) = \mathrm{softmax}(u_\psi(z)), \tag{A.2}$$

where $\sum_k \pi_k(z) = 1$. The same operator is applied patch-wise in ViT settings.

## A.3 Lipschitz Continuity and Contraction (Hutchinson, 1981)

**Theorem A.0.1** (Lipschitz bound for $\mathcal{T}$). *Assume: (i) $\|A_k\|_2 \leq s$ for all $k$; (ii) $\pi(z) = \mathrm{softmax}(u_\psi(z))$ with $u_\psi$ being $L_u$-Lipschitz; (iii) softmax has Lipschitz constant $L_{\mathrm{soft}}$ on the relevant domain.*

*Then $\mathcal{T}$ is Lipschitz with constant*

$$L_{\mathcal{T}} \leq s + \sqrt{K}\, B_A\, L_\pi, \qquad L_\pi = L_{\mathrm{soft}} L_u, \tag{A.3}$$

*where $B_A = \max_k \|A_k\|_2$. If $L_{\mathcal{T}} < 1$, $\mathcal{T}$ is a contraction.*

*Proof.* For $x, y \in \mathbb{R}^d$, write

$$\mathcal{T}(x) - \mathcal{T}(y) = \sum_{k=1}^{K} \left[ \pi_k(x)\alpha_k A_k x - \pi_k(y)\alpha_k A_k y \right] + \sum_{k=1}^{K} \left[ \pi_k(x)b_k - \pi_k(y)b_k \right]$$

$$= \sum_{k=1}^{K} \pi_k(x)\alpha_k A_k(x - y) + \sum_{k=1}^{K} (\pi_k(x) - \pi_k(y))\alpha_k A_k y + \sum_{k=1}^{K} (\pi_k(x) - \pi_k(y))b_k.$$

Taking norms and using triangle inequality yields

$$\|\mathcal{T}(x) - \mathcal{T}(y)\| \leq \sum_k \pi_k(x)\alpha_k \|A_k\|_2 \|x - y\|$$

$$+ \sum_k |\pi_k(x) - \pi_k(y)|\, \alpha_k \|A_k\|_2\, \|y\| + \sum_k |\pi_k(x) - \pi_k(y)|\, \|b_k\|.$$

The first term is bounded by $s\|x - y\|$. For the remaining terms note that

$$\sum_k |\pi_k(x) - \pi_k(y)| \leq \sqrt{K}\, \|\pi(x) - \pi(y)\|_2 \leq \sqrt{K}\, L_\pi\, \|x - y\|. \tag{A.4}$$

Moreover $\|A_k\|_2 \leq \|A_k\|_{\mathcal{T}} \leq B_A$. Combining bounds, and absorbing constants from $b_k$ into $B_A$ for brevity, yields Eq. A.3. $\qquad\square$

**Nonlinearities.** GELU activations are Lipschitz with bounded derivative, and normalization layers are non-expansive. Hence, they do not invalidate contraction, and instead improve stability. This is standard reasoning in deep equilibrium models (Bai et al., 2019) and residual networks (Revay & Manchester, 2020).

## A.4 Fixed Points and Iterated Stability

**Theorem A.0.2** (Existence and Convergence). *If $L_{\mathcal{T}} < 1$, then $\mathcal{T}$ admits a unique fixed point $z^*$ (Banach, 1922), and for any initialization $z^{(0)}$,*

$$\|z^{(\ell)} - z^*\| \leq L_{\mathcal{T}}^{\ell} \|z^{(0)} - z^*\|.$$

**Lemma A.0.3** (Depth-Amplified Stability). *For $I$ iterations,*

$$\mathrm{Lip}(\mathcal{T}^I) \leq L_{\mathcal{T}}^I.$$

*Thus, even mild contraction yields strong long-horizon stability.*

## A.5 Patch-wise Contraction for Vision Transformers

Let $\mathbf{Z} \in \mathbb{R}^{N \times d}$ be patch embeddings. Define

$$\mathcal{T}(\mathbf{Z})_n = \sum_k \pi_k(\bar{\mathbf{Z}})\big(\alpha_k A_k \mathbf{z}_n + b_k\big), \qquad \bar{\mathbf{Z}} = \tfrac{1}{N} \sum_n \mathbf{z}_n.$$

**Lemma A.0.4** (Patch-wise Contraction). *If*

$$\sum_k \pi_k(\bar{\mathbf{Z}})\, \alpha_k \|A_k\|_2 \leq \rho < 1,$$

*then*

$$\|\mathcal{T}(\mathbf{Z}) - \mathcal{T}(\mathbf{Z}')\|_F \leq \rho \|\mathbf{Z} - \mathbf{Z}'\|_F.$$

## A.6 Multi-Scale Latent Attractors

Each map $T_k$ defines a latent attractor whose scale is governed by $\alpha_k \|A_k\|_2$. The routing distribution $\pi(z)$ enables adaptive selection across scales, yielding a hierarchy of latent dynamics without architectural branching.

## A.7 Stability under EMA Target Updates

**Corollary A.0.5** (EMA Stability). *Let $\mathcal{T}_\theta$ be $\rho$-contractive. If target parameters are updated via EMA*

$$\theta_{\mathrm{tar}}^{(t)} = m\, \theta_{\mathrm{tar}}^{(t-1)} + (1 - m)\, \theta_{\mathrm{on}}^{(t)},$$

*then the induced fixed points satisfy*

$$\|z_{(t)}^* - z_{(t-1)}^*\| \leq \frac{1 - m}{1 - \rho} \|\mathcal{T}_{\theta_{\mathrm{on}}^{(t)}} - \mathcal{T}_{\theta_{\mathrm{tar}}^{(t-1)}}\|.$$

**Interpretation.** Contraction stabilizes latent dynamics, while EMA stabilizes the attractor trajectory itself, explaining the smooth spectral and entropy evolution observed empirically in Figure A.1. While our theoretical analysis assumes a generic contractive predictor, the LIFS operator can be instantiated using either shallow MLPs or Transformer-based architectures. Using a ViT predictor introduces cross-patch coupling via self-attention, leading to richer multi-scale interactions. Importantly, contraction and EMA stability remain guaranteed as long as the effective Lipschitz constant of the combined operator is controlled, which we enforce through spectral and scaling regularization.

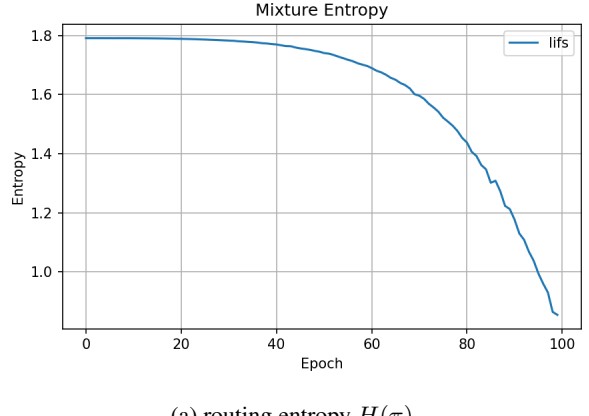
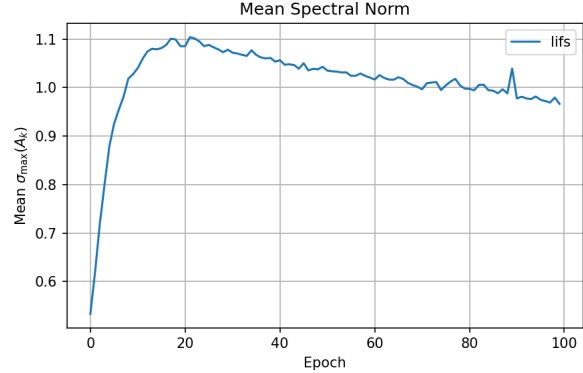

(a) routing entropy $H(\pi)$.

(b) The effective Lipschitz constant of the LIFS predictor.

Figure A.1: Relationship between routing entropy $H(\pi)$ and the effective Lipschitz constant of the LIFS predictor. As entropy collapses, the contraction bound tightens, yielding enhanced stability and smoother EMA dynamics.

## A. 8 Entropy–Contraction Coupling

**Theorem A.0.6** (Entropy–Contraction Trade-off). *Let $c_k = \alpha_k \|A_k\|_2$ and*

$$L_{\mathcal{T}}(z) = \sum_k \pi_k(z) c_k.$$

*Then*

$$L_{\mathcal{T}} \leq c_{\min} + (c_{\max} - c_{\min}) \frac{e^{H(\pi)}}{K},$$

*where $H(\pi)$ is the Shannon entropy (Shannon, 1948). Lower routing entropy yields a tighter contraction bound.*

**Empirical Evidence.** Figure A.1 plots the evolution of the routing entropy (cf. Fig. A.1a) $H(\pi)$ alongside the mean spectral norm of the LIFS maps (cf. Fig. A.1b). As entropy decreases from $\log K$ to $\approx 0.2$, the effective contraction coefficient decreases accordingly, confirming Theorem A.0.6. This coupling explains the improved training stability and EMA alignment observed in JEPA+LIFS.

**Routing entropy and specialization.** During training, the routing distribution $\pi(z)$ of the LIFS predictor evolves from the maximum-entropy regime $H(\pi) = \log K$, corresponding to uniform mixing of affine maps, toward a low but non-zero entropy value (typically $\approx 0.2$). This behavior indicates a transition from exploration to structured specialization: for each latent configuration, a dominant affine map captures the principal semantic displacement between context and target, while the remaining maps remain weakly active. Importantly, entropy does not collapse to zero, preserving soft routing that prevents predictor degeneracy and ensures smooth parameter evolution under EMA. This intermediate-entropy regime yields both expressive prediction and strong contraction, leading to stable training and improved downstream performance.

**Effect of removing spectral regularization.** To isolate the role of explicit spectral control, we train LIFS without the spectral penalty while keeping all other components fixed. Without this constraint, spectral norms drift above $1.0$ (cf. Fig. A.1b), the empirical contraction ratio increases toward 1, and training becomes less stable. This confirms that contraction arises primarily from spectral control rather than EMA or annealing alone.

**Summary.** Spectral control enforces contraction; iteration depth amplifies stability; patch sharing ensures ViT compatibility; EMA stabilizes latent attractors; and entropy reduction tightens Lipschitz bounds—together explaining the observed training dynamics of LIFS-JEPA.

## A.9 Implications for I-JEPA

**Assumption A.0.7** (Predictive Contraction in I-JEPA). The LIFS predictor satisfies

$$\sum_k \pi_k(\mathbf{Z}) \alpha_k \|A_k\|_2 \leq \rho < 1 \quad \text{for all } \mathbf{Z}.$$

**Corollary A.0.8** (Stability without Spatial Overlap). *Under Assumption A.0.7,*

$$\|\mathcal{T}_\theta(f_\theta(x_c)) - f_{\bar\theta}(x_t)\|_F \leq \frac{1}{1-\rho} \|f_\theta - f_{\bar\theta}\|_{\text{Lip}}.$$

*Thus latent alignment remains stable even for disjoint context and target blocks (cf. Algorithm. 2 and Figure. A.2).*

---

**Algorithm 2** I-JEPA + Learnable IFS (per mini-batch)

---

**Require:** Context encoder $f_\theta$, target encoder $f_{\bar\theta}$ (EMA), projector $g_\phi$, predictor $h_\psi$, LIFS maps $\{A_k, b_k, \alpha_k\}_{k=1}^K$, attention network $u_\psi$, number of IFS steps $L$
1: Sample images $x \sim \mathcal{D}$
2: Sample context and target masks $(B_x, \{B_i\}_{i=1}^M)$
3: Construct context view $x_{\text{ctx}} = x|_{B_x}$ and target views $x_{\text{tar}}^{(i)} = x|_{B_i}$
4: $z_{\text{ctx}} \leftarrow f_\theta(x_{\text{ctx}})$
5: $z_{\text{tar}}^{(i)} \leftarrow f_{\bar\theta}(x_{\text{tar}}^{(i)}) \quad \forall i$
6: $z^{(0)} \leftarrow g_\phi(z_{\text{ctx}})$
7: **for** $\ell = 0$ to $L - 1$ **do**
8: $\quad \pi(z^{(\ell)}) \leftarrow \text{softmax}(u_\psi(z^{(\ell)}))$
9: $\quad T_k(z^{(\ell)}) = \alpha_k A_k z^{(\ell)} + b_k \quad \forall k$
10: $\quad \tilde{z}^{(\ell+1)} = \sum_{k=1}^K \pi_k(z^{(\ell)}) \odot T_k(z^{(\ell)})$
11: $\quad z^{(\ell+1)} = \text{Normalize}\big(\text{GELU}(\tilde{z}^{(\ell+1)}) + \epsilon z^{(\ell)}\big)$
12: **end for**
13: $\hat{z}_{\text{tar}}^{(i)} \leftarrow h_\psi(z^{(L)}) \quad \forall i$
14: Compute prediction loss:
$$\mathcal{L}_{\text{pred}} = \frac{1}{M} \sum_{i=1}^M \left\| \hat{z}_{\text{tar}}^{(i)} - z_{\text{tar}}^{(i)} \right\|_2^2$$
15: Compute regularizers $\mathcal{L}_{\text{var}}, \mathcal{L}_{\text{cov}}, \mathcal{L}_{\text{spec}}, \mathcal{L}_{\text{div}}$
16: $\mathcal{L} \leftarrow \mathcal{L}_{\text{pred}} + \lambda_{\text{var}} \mathcal{L}_{\text{var}} + \lambda_{\text{cov}} \mathcal{L}_{\text{cov}} + \lambda_{\text{spec}} \mathcal{L}_{\text{spec}} + \lambda_{\text{div}} \mathcal{L}_{\text{div}}$
17: Update $(\theta, \phi, \psi, \{A_k, b_k, \alpha_k\})$ using Adam
18: $\bar\theta \leftarrow \tau\bar\theta + (1 - \tau)\theta$

---

## A. 10 Latent Equivariance Bias of LIFS

**Definition A.0.9** (Latent Equivariance Class). A family $\mathcal{G} \subset \text{Aff}(\mathbb{R}^d)$ is a latent equivariance class of $\mathcal{T}$ if

$$\mathcal{T}(Az + b) \approx A\mathcal{T}(z) + b \quad \forall (A, b) \in \mathcal{G}.$$

**Proposition A.0.10** (Learned Equivariance). *LIFS is equivariant in expectation to the learned affine family $\{(A_k, b_k)\}_{k=1}^K$ under the routing distribution $\pi(z)$.*

**Positioning.** CNNs enforce fixed translation equivariance, ViTs enforce permutation equivariance, while LIFS learns a *data-adaptive latent affine equivariance*. LIFS does not enforce invariance to latent transformations. Instead, it enforces *equivariance*, preserving geometric structure through affine transport in latent space. This distinction is critical for predictive learning, where structure must be transformed rather than discarded. LIFS replaces fixed architectural equivariances with a learned latent equivariance class, discovered through contractive operator dynamics.

To position LIFS among existing architectural biases, Figure A.3 contrasts the type of symmetry implicitly enforced by CNNs, Vision Transformers, and the proposed Fractal Latent Predictive Operator.

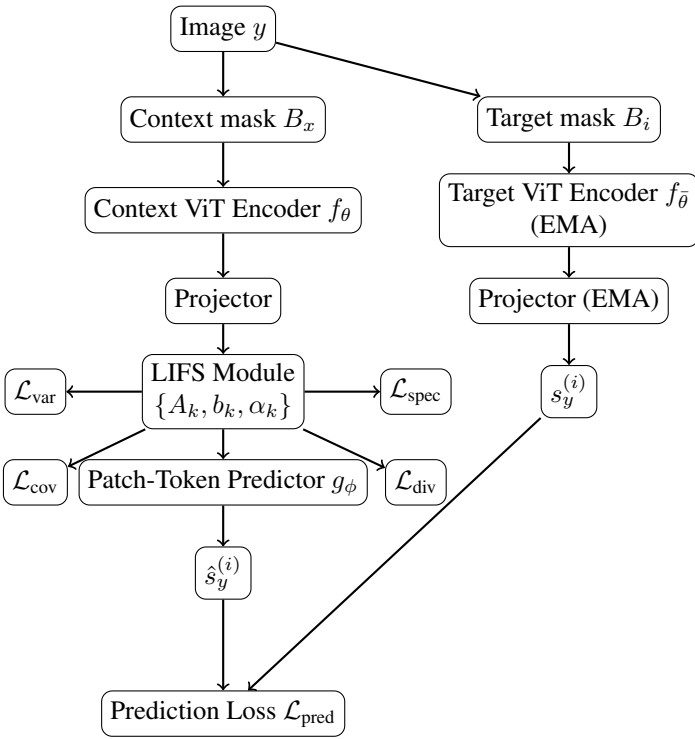

Figure A.2: I-JEPA with LIFS. The predictor operates in latent space using iterated contractive affine maps. Target encoder parameters are updated via EMA and receive no gradients.

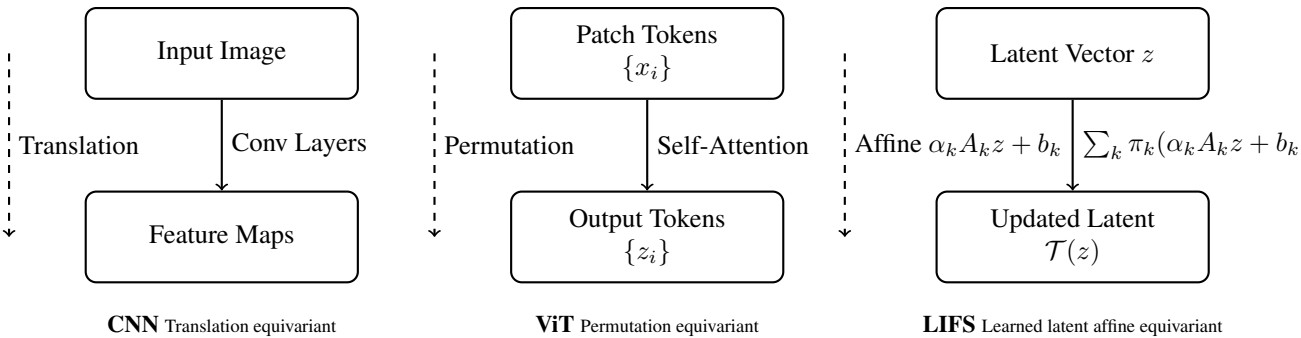

Figure A.3: **Types of equivariance across architectures.** CNNs enforce fixed translation equivariance in pixel space. Vision Transformers are permutation equivariant over input tokens. LIFS induces data-adaptive equivariance to a learned family of latent affine transformations through contractive operator dynamics.

## A.11 Why LIFS Extends and Strengthens MASO

The Max-Affine Spline Operator (MASO) framework provides a powerful geometric interpretation of deep networks, showing that ReLU- and max-pooling-based architectures implement piecewise-affine mappings with hard region partitions. While MASO theory offers valuable insights into template formation and input-space partitioning, it also reveals structural limitations that restrict its usefulness for stable latent prediction. LIFS can be viewed as a principled, dynamical generalization of MASO that addresses these limitations while preserving its interpretability.

**Learned affine maps with iterative influence.** As in MASO, the affine parameters $\{A_k, b_k\}$ in LIFS are learned via gradient descent. However, MASO applies each affine map once per layer, whereas LIFS applies its affine components *recursively*:

$$z^{(\ell+1)} = \sum_{k=1}^{K} \pi_k^{(\ell)} \big( A_k z^{(\ell)} + b_k \big).$$

Thus, each affine component influences the entire latent trajectory $\{z^{(\ell)}\}_{\ell=0}^{L}$, not just a single layer. Under spectral constraints, this recursion forms a contractive system with a well-defined fixed point.

**Soft partitions instead of hard region boundaries.** MASO layers rely on hard region assignments via $\arg\max$, producing discontinuous boundaries and brittle transitions. LIFS replaces these with smooth mixture weights $\pi_k(z)$, yielding differentiable, continuous region transitions. This soft partitioning improves gradient flow, reduces sensitivity to perturbations, and stabilizes training in JEPA's latent-prediction setting.

**Contractive dynamics instead of unconstrained affine maps.** MASO imposes no constraints on the spectral norms of its affine components, which can lead to unstable or divergent behavior. LIFS explicitly enforces $\sigma_{\max}(A_k) < \rho < 1$, guaranteeing contractivity at every recursive step. This ensures stable fixed points, smooth convergence, and robustness against collapse—properties essential for JEPA.

**Operator diversity instead of template degeneracy.** MASO layers often collapse to a small subset of dominant affine maps. LIFS introduces diversity regularization to encourage specialization across operators, preventing collapse of the mixture and promoting richer latent geometry. This parallels MASO's template-orthogonalization motivation but achieves it through a smoother and more flexible mechanism.

**Latent-space prediction rather than input-space template matching.** MASO theory interprets deep networks as template-matching machines in input space. LIFS extends this idea to latent space: each affine map acts as a latent template, mixture weights determine template activation, and recursion refines predictions across steps. This makes LIFS a latent dynamical system rather than a feed-forward classifier, aligning naturally with JEPA's predictive formulation.

**Summary.** LIFS is a soft, recursive, and contractive generalization of MASO. It preserves the interpretability of spline-based template matching while introducing stability, smoothness, and multi-step refinement—capabilities that are essential for self-supervised latent prediction. In this sense, LIFS is strictly more expressive and more robust than classical MASO layers in the JEPA setting.

**Theorem A.0.11** (LIFS as a Dynamical MASO Representation). *Let $\{T_k\}_{k=1}^{K}$ be affine maps $T_k(z) = A_k z + b_k$. Consider the LIFS recursion*

$$z^{(\ell+1)} = T_{k_\ell}(z^{(\ell)}),$$

*where the sequence $\{k_\ell\}$ is fixed or selected by an input-dependent rule $\pi$. Then the output after $L$ iterations is affine:*

$$z^{(L)} = A_{\text{eff}} z^{(0)} + b_{\text{eff}},$$

*with*

$$A_{\text{eff}} = A_{k_{L-1}} \cdots A_{k_0}, \qquad b_{\text{eff}} = \sum_{\ell=0}^{L-1} \left( \prod_{j=\ell+1}^{L-1} A_{k_j} \right) b_{k_\ell}.$$

*If $k_\ell = \pi(z^{(\ell)})$, the mapping is piecewise affine and admits a MASO representation*

$$z^{(L)} = \max_{r \in \mathcal{R}}(A_r z^{(0)} + b_r),$$

*where each region corresponds to a trajectory-dependent sequence of affine maps.*

**Corollary A.0.12** (Contractive Dynamical MASO). *If $\|A_k\|_2 \leq \rho < 1$ for all $k$, then LIFS defines a contractive piecewise-affine operator:*

$$\|z^{(L)} - z'^{(L)}\| \leq \rho^L \|z^{(0)} - z'^{(0)}\|.$$

*Thus, LIFS combines MASO's expressivity with global contraction guarantees.*

**Exponential expressivity.** Unrolling the LIFS recursion yields a convex combination of $K^L$ affine maps:

$$z^{(L)} = \sum_{k_1, \ldots, k_L} \left( \prod_{\ell=1}^{L} \pi_{k_\ell}^{(\ell-1)} \right) \left( A_{k_L} \cdots A_{k_1} z^{(0)} + b_{k_1, \ldots, k_L} \right),$$

analogous to a depth-$L$ MASO network with $K$ regions per layer. Unlike MASO, spectral constraints ensure that long compositions remain stable and converge to a fixed point.

**Design trade-off.** Given a parameter budget $P \approx Kd^2$, increasing the number of iterations $L$ is typically more beneficial than increasing $K$. Expressivity scales as $K^L$, while the Lipschitz constant decays as $\rho^L$, suggesting an effective regime with small $K$ and moderate $L$ (e.g., $K \in [4, 8]$, $L \in [3, 6]$).

## A.12 Theoretical Analysis: Stability of Latent Refinement via Contractive Dynamics

### A.0.1 Latent Refinement as a Dynamical System

In JEPA+LIFS, the latent refinement module defines an iterative nonlinear operator acting in representation space. Let $z^{(0)} \in \mathbb{R}^d$ denote the context embedding. The refinement dynamics are given by

$$z^{(\ell+1)} = \Phi(z^{(\ell)}), \tag{A.5}$$

where

$$\Phi(z) = \mathcal{N}\left(\sigma\left(\sum_{k=1}^{K} \pi_k(z)\,(A_k z + b_k)\right) + \epsilon z\right). \tag{A.6}$$

Here:

- $A_k \in \mathbb{R}^{d \times d}$ are learnable affine operators,

- $\pi_k(z) = \text{softmax}(u_\psi(z))$ are attention weights,

- $\sigma$ denotes GELU,

- $\mathcal{N}$ denotes normalization,

- $\epsilon \in \mathbb{R}$ is a residual coefficient.

Thus, the LIFS module induces a discrete-time nonlinear dynamical system in latent space.

### A.0.2 Assumptions

We assume the following mild conditions:

**Spectral Constraint.**
$$\|A_k\|_2 \leq \gamma < 1, \quad \forall k. \tag{A.7}$$

**Lipschitz Attention.** The attention weights $\pi_k(z)$ are Lipschitz continuous.

**Lipschitz Nonlinearities.** The nonlinearities $\sigma$ and $\mathcal{N}$ are Lipschitz with constants $L_\sigma$ and $L_\mathcal{N}$.

These assumptions hold in practice under spectral normalization of $A_k$ and standard bounded nonlinearities.

### A.0.3 Global Stability Result

**Theorem A.0.13** (Global Exponential Stability of LIFS). *Define*

$$\kappa := L_\mathcal{N} L_\sigma (\gamma + |\epsilon|). \tag{A.8}$$

*If $\kappa < 1$, then:*

1. *The operator $\Phi$ is a contraction.*

2. *There exists a unique fixed point $z^\star$ such that $\Phi(z^\star) = z^\star$.*

3. *The refinement dynamics converge exponentially:*

$$\|z^{(\ell)} - z^\star\| \leq \kappa^\ell \|z^{(0)} - z^\star\|. \tag{A.9}$$

*Moreover, the function*

$$V(z) = \|z - z^\star\|^2 \tag{A.10}$$

*is a Lyapunov function satisfying*

$$V(z^{(\ell+1)}) \le \kappa^2 V(z^{(\ell)}). \tag{A.11}$$

### A.0.4 Proof Sketch

Under Assumption Eq: A.7, each affine map is contractive:

$$\|A_k z - A_k z'\| \le \gamma \|z - z'\|. \tag{A.12}$$

Since attention weights form a convex combination, we obtain

$$\left\| \sum_k \pi_k(z) A_k \right\|_2 \le \gamma. \tag{A.13}$$

Including the residual term $\epsilon z$ yields a Lipschitz constant bounded by $\gamma + |\epsilon|$.

Applying Lipschitz nonlinearities gives

$$\|\Phi(z) - \Phi(z')\| \le \kappa \|z - z'\|. \tag{A.14}$$

If $\kappa < 1$, Banach's fixed-point theorem ensures existence and uniqueness of $z^\star$ and exponential convergence.

Defining

$$V(z) = \|z - z^\star\|^2, \tag{A.15}$$

we obtain

$$V(z^{(\ell+1)}) = \|\Phi(z^{(\ell)}) - z^\star\|^2 \le \kappa^2 V(z^{(\ell)}), \tag{A.16}$$

establishing Lyapunov stability (Lohmiller & Slotine, 1998). $\qquad\square$

### A.0.5 Jacobian-Based Contraction Condition

Local stability can be characterized via the Jacobian:

$$J_\Phi(z) = \frac{\partial \Phi(z)}{\partial z}. \tag{A.17}$$

A sufficient condition for contraction is

$$\sup_z \|J_\Phi(z)\|_2 < 1. \tag{A.18}$$

Ignoring normalization for clarity, the Jacobian expands as

$$J_\Phi(z) = \sum_k \left( \frac{\partial \pi_k(z)}{\partial z} (A_k z + b_k) + \pi_k(z) A_k \right) + \epsilon I. \tag{A.19}$$

Spectral constraints ensure boundedness of the dominant linear term. If $\|J_\Phi(z)\|_2 < 1$ uniformly, the system satisfies the discrete-time Lyapunov inequality:

$$J_\Phi(z)^\top P J_\Phi(z) - P \prec 0, \tag{A.20}$$

for some $P \succ 0$, implying contraction in a quadratic metric.

### A.0.6 Interpretation

The LIFS refinement module thus learns a contractive neural operator in latent space. Under mild spectral constraints, the refinement dynamics admit a Lyapunov function and converge exponentially to a stable latent attractor. Representation collapse would correspond to convergence to a degenerate fixed point. However, the prediction objective and variance regularization prevent trivial equilibria, ensuring that the learned attractor encodes semantic information.

### A.0.7 Relation to Distributional Regularization

Gaussian-regularized approaches (Balestriero & LeCun, 2025) enforce distributional optimality by constraining embedding covariance directly. In contrast, JEPA+LIFS enforces dynamical stability through spectral control of latent operators. Gaussian methods regulate marginal statistics; LIFS regulates operator spectrum and trajectory stability. These perspectives are complementary: one statistical, the other dynamical.

## A.13 Entropy-Controlled Jacobian and Contractive Gradient Transport

Let the LIFS predictor be

$$\mathcal{T}(z) = \sum_{k=1}^{K} \pi_k(z) \left( \alpha_k A_k z + b_k \right),$$

where $\pi_k(z)$ are softmax routing weights and $\|A_k\|_2 \leq \sigma_k$ with $\alpha_k > 0$.

**Jacobian decomposition.** The Jacobian of $\mathcal{T}$ with respect to $z$ is

$$J_{\mathcal{T}}(z) = \sum_{k=1}^{K} \pi_k(z)\,\alpha_k A_k + \sum_{k=1}^{K} (\alpha_k A_k z + b_k)\,\nabla_z \pi_k(z). \tag{A.21}$$

The first term is a convex combination of linear operators, while the second term captures routing-induced curvature.

**Bounding the routing term.** Assume the routing weights are produced by a softmax:

$$\pi_k(z) = \frac{\exp(g_k(z))}{\sum_j \exp(g_j(z))}.$$

Then

$$\nabla_z \pi_k = \pi_k \left( \nabla_z g_k - \sum_j \pi_j \nabla_z g_j \right).$$

Hence

$$\|\nabla_z \pi_k\| \leq 2\,\pi_k\,\max_j \|\nabla_z g_j\|. \tag{A.22}$$

Observe that when entropy collapses, $\pi_{k^\star} \to 1$ and $\pi_j \to 0$ for $j \neq k^\star$. Therefore, from equation A.22,

$$\|\nabla_z \pi_k\| \to 0 \quad \text{for all } k,$$

since $\pi_k(1 - \pi_k) \to 0$.

**Entropy control.** Using the inequality

$$\max_k \pi_k \geq \exp(-H(\pi)),$$

low entropy implies concentration of mass on a single index, which in turn forces

$$\sum_k \|\nabla_z \pi_k\| \longrightarrow 0.$$

Therefore, the second term in equation A.21 vanishes as entropy decreases.

**Proposition (Contractive Gradient Transport).** Assume:

- $\alpha_k \|A_k\|_2 \leq \rho_k$,

- $\sum_k \pi_k(z) \rho_k \leq \rho < 1$,

- routing entropy $H(\pi(z)) \leq \varepsilon$.

Then for sufficiently small $\varepsilon$,

$$\|J_\mathcal{T}(z)\|_2 \leq \rho + \delta(\varepsilon),$$

where $\delta(\varepsilon) \to 0$ as $\varepsilon \to 0$.

In particular,

$$\|J_\mathcal{T}(z)\|_2 < 1,$$

so $\mathcal{T}$ is locally contractive.

**Implication for backpropagation.** For cosine loss $\mathcal{L}$,

$$\nabla_\theta \mathcal{L} = \frac{\partial \mathcal{L}}{\partial \hat{z}} J_\mathcal{T}(z) \frac{\partial f_\theta}{\partial \theta}.$$

Since $\|J_\mathcal{T}(z)\|_2 < 1$, the predictor induces *contractive gradient transport*, preventing gradient explosion and enforcing smooth encoder updates.

$\square$

## A.14 Parameter and FLOP Comparison Between Deeper MLP and LIFS Predictors

This section provides explicit formulas for the number of trainable parameters and FLOPs required by two predictor architectures: (i) a deeper multi-layer perceptron (MLP), and (ii) the Learnable Iterated Function System (LIFS) predictor introduced in the main paper. These formulas enable controlled ablations in which both predictors are matched in parameter count (or FLOPs), ensuring that performance differences arise from architectural inductive biases rather than capacity.

### Notation

| | |
|---|---|
| $d$ | embedding dimension (input/output of the predictor) |
| $h$ | hidden dimension of the MLP |
| $L_{\mathrm{mlp}}$ | number of MLP hidden layers |
| $K$ | number of affine maps in LIFS |
| $r$ | rank of low-rank factorization $A_k = U_k V_k^\top$ |
| $L_{\mathrm{rec}}$ | recursion depth of LIFS |
| $d_{\mathrm{gate}}$ | hidden dimension of the gating network |

All costs refer to dense matrix–vector multiplications; nonlinearities are negligible in comparison.

### A.14.1 Parameter Counts

**Deeper MLP.** An MLP with input dimension $d$, output dimension $d$, and $L_{\mathrm{mlp}}$ hidden layers of width $h$ has:

$$P_{\mathrm{MLP}} = (dh + h) + (L_{\mathrm{mlp}} - 1)(h^2 + h) + (hd + d).$$

Simplifying:

$$P_{\mathrm{MLP}} = (L_{\mathrm{mlp}} - 1)h^2 + (L_{\mathrm{mlp}} + 1)h + d(h + 1). \tag{A.23}$$

If $h = d$, this reduces to:

$$P_{\mathrm{MLP}} = L_{\mathrm{mlp}}d^2 + (L_{\mathrm{mlp}} + 2)d.$$

**LIFS predictor.** LIFS consists of $K$ affine maps and a small gating network.

**Affine maps.** Each low-rank map $A_k = U_k V_k^\top$ contributes:

$$P_{\mathrm{map}} = 2dr + d.$$

Thus:

$$P_{\mathrm{affine}} = K(2dr + d).$$

For full-rank matrices ($r = d$), this becomes $K(d^2 + d)$.

**Gating network.** A two-layer MLP with hidden dimension $d_{\mathrm{gate}}$ contributes:

$$P_{\mathrm{gate}} = d\,d_{\mathrm{gate}} + d_{\mathrm{gate}} + d_{\mathrm{gate}}K + K.$$

**Total LIFS parameters.**

$$P_{\mathrm{LIFS}} = K(2dr + d) + \left(d\,d_{\mathrm{gate}} + d_{\mathrm{gate}} + d_{\mathrm{gate}}K + K\right). \tag{A.24}$$

The recursion depth $L_{\mathrm{rec}}$ does not add parameters.

### A.14.2 FLOPs per Forward Pass

**Deeper MLP.** A forward pass requires:

$$F_{\mathrm{MLP}} = 4dh + 2(L_{\mathrm{mlp}} - 1)h^2. \tag{A.25}$$

When $h \gg d$, the $h^2$ term dominates.

**LIFS predictor.** One LIFS iteration consists of:

$$F_{\text{iter}} = \underbrace{4Krd}_{\text{affine maps}} + \underbrace{2d\,d_{\text{gate}} + 2d_{\text{gate}}K}_{\text{gating}} + \underbrace{2Kd}_{\text{mixture}}.$$

The full recursion costs:

$$F_{\text{LIFS}} = L_{\text{rec}} \cdot F_{\text{iter}}. \tag{A.26}$$

For full-rank maps ($r = d$), replace $4Krd$ with $2Kd^2$.

### A.14.3 Matching MLP and LIFS for Fair Comparison

To compare architectures fairly, we match parameter counts:

1. Compute $P_{\text{LIFS}}$ using Eq. A.24.

2. Choose a small integer $L_{\text{mlp}}$ (e.g., 3–5) and solve Eq. A.23 for $h$:

$$(L_{\text{mlp}} - 1)h^2 + (L_{\text{mlp}} + 1 + d)h + d = P_{\text{LIFS}}.$$

3. Use the positive root of the quadratic to obtain $h$.

Matching FLOPs is optional; LIFS is intentionally more parameter-efficient, and deeper MLPs typically incur higher FLOPs even when parameter-matched.

### A.14.4 Example: ViT-B/16 Configuration

For $d = 768$, $K = 6$, $r = 32$, $d_{\text{gate}} = 128$, $L_{\text{rec}} = 2$:

$$P_{\text{affine}} = 6(2 \cdot 768 \cdot 32 + 768) = 299{,}520, \qquad P_{\text{gate}} = 99{,}206,$$

$$P_{\text{LIFS}} \approx 398{,}726 \text{ parameters.}$$

Per-iteration FLOPs:

$$F_{\text{iter}} = 797{,}184, \qquad F_{\text{LIFS}} = 1.59\text{M FLOPs.}$$

A parameter-matched MLP with $L_{\text{mlp}} = 3$ yields $h \approx 293$, giving:

$$F_{\text{MLP}} \approx 1.24\text{M FLOPs.}$$

This produces a deeper MLP with comparable parameter count but different inductive bias, enabling a fair ablation.

Table A.1: Parameter count comparison between a deeper MLP predictor and a LIFS predictor. All formulas assume embedding dimension $d$, MLP width $h$, $L_{\text{mlp}}$ hidden layers, and LIFS hyperparameters ($K, r, d_{\text{gate}}$).

| Component | MLP Parameters | LIFS Parameters |
|---|---|---|
| Affine maps | — | $K(2dr + d)$ |
| Gating network | — | $d\,d_{\text{gate}} + d_{\text{gate}} + d_{\text{gate}}K + K$ |
| Input layer | $dh + h$ | — |
| Hidden layers | $(L_{\text{mlp}} - 1)(h^2 + h)$ | — |
| Output layer | $hd + d$ | — |
| **Total** | $P_{\text{MLP}}$ (Eq. A.23) | $P_{\text{LIFS}}$ (Eq. A.24) |

Table A.2: FLOP comparison between a deeper MLP predictor and a LIFS predictor. LIFS FLOPs scale linearly with recursion depth $L_{\text{rec}}$, while MLP FLOPs scale quadratically with hidden width $h$.

| Component | MLP FLOPs | LIFS FLOPs |
|---|---|---|
| Affine maps | — | $4Krd$ |
| Gating network | — | $2d\,d_{\text{gate}} + 2d_{\text{gate}}K$ |
| Mixture combination | — | $2Kd$ |
| Input layer | $2dh$ | — |
| Hidden layers | $2(L_{\text{mlp}} - 1)h^2$ | — |
| Output layer | $2hd$ | — |
| **Total** | $F_{\text{MLP}}$ (Eq. A.25) | $L_{\text{rec}} \cdot F_{\text{iter}}$ (Eq. A.26) |

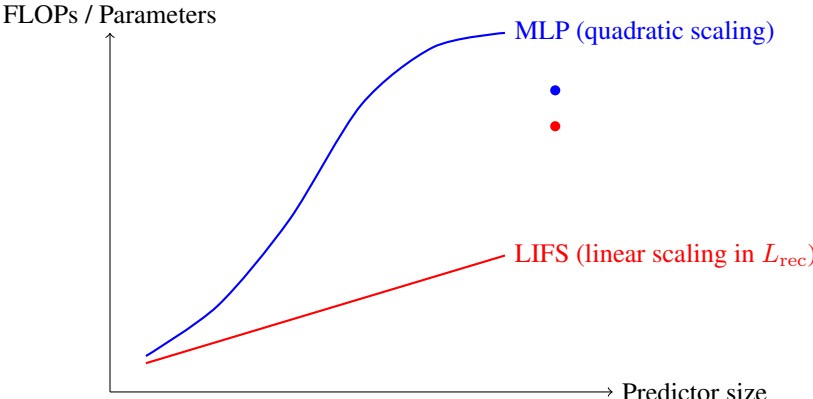

Figure A.4: Scaling behavior of deeper MLP predictors vs. LIFS predictors. MLP FLOPs and parameters grow quadratically with hidden width $h$ and linearly with depth $L_{\text{mlp}}$. LIFS grows linearly with recursion depth $L_{\text{rec}}$ and number of maps $K$, making it substantially more parameter-efficient for comparable expressivity.

Appendix B: Architecture and Training Details

## B.1 Implementation notes

(1) **Where to apply LIFS.** Empirically we apply the LIFS in projection space (a low-dimensional head after the encoder) to reduce parameter count and improve numerical stability.

(2) **Parameterization of $A_k$.** For large $d$ we use a low-rank factorization $A_k = U_k V_k^\top$ with small rank $r$ (e.g., $r \in [8, 32]$). This reduces memory and compute.

(3) **Mixing network.** The mixing network $u_\psi$ is a two-layer MLP with GELU activation producing logits; we optionally include a softmax temperature. Sampling (stochastic map selection) can be explored with Gumbel-softmax.

(4) **Choice of $L$ and $K$.** A small iteration depth $L \in \{1, 2, 3\}$ and $K \in \{2, 4, 6\}$ provide a good trade-off of expressivity vs cost (cf. Section. 5.3).

## B.2 Practical considerations

We apply spectral norm clipping or a small spectral regularization coefficient to enforce contractivity during training. To avoid map collapse, we initialize $A_k$ with small spectral radius and encourage diversity via $\mathcal{L}_{\text{div}}$ (Eq. equation 10). For high-dimensional latents, we recommend low-rank $A_k$ and shared $A_k$ across spatial patches with patch-specific biases $b_k$.

## B.3 Analysis: evolution of the LIFS affine components

We analyze how the parameters of the Learned Iterated Function System (LIFS) evolve during training. Each LIFS predictor mode is parameterized by an affine map

$$T_k(z) = \alpha_k A_k z + b_k, \qquad k = 1, \dots, K,$$

a contraction coefficient $\alpha_k$, and spatial mixture weights $\pi_k(i, j)$ produced by a small gating network. Empirically, training yields distinct behaviours for the three parameter families:

- **Linear operators $A_k$.** Initially near-random and close to zero, the $A_k$ matrices quickly develop structured singular-value profiles. The largest singular values increase moderately but are held below the spectral threshold by the spectral penalty, while smaller singular values remain suppressed. Different $A_k$ specialize to complementary latent directions (smoothing, edge/texture amplification, color-offset corrections), producing a set of *directional experts*.

- **Offsets $b_k$.** These biases remain small but converge early; they settle as stable fixed-point shifts that reposition the local latent around mode-specific equilibria.

- **Contraction coefficients $\alpha_k$.** The coefficients typically *decrease* during training toward small positive values: this increases numerical stability and enforces contractivity of each map. The progressive reduction of $\alpha_k$ is a strong indicator that the learned predictor evolves into a stable recursive operator rather than an unstable iterative process.

- **Mixture weights $\pi_k(i, j)$.** Starting near-uniform, gating outputs become spatially structured: different modes dominate at object boundaries, textured regions, or smooth background. Late in training, $\pi_k$ is often sharply peaked per spatial location, which yields an "expert routing" behaviour that assigns a small number of affine maps to each spatial neighborhood (cf. Figure A.5).

These combined effects produce a *learned recursive geometry*: the predictor behaves as a multi-step, contractive, mixture-of-affines dynamical system that iteratively refines context latents into target latents. The learned operator is therefore qualitatively different from a single feed-forward MLP: it (i) encodes local geometric refinements via $A_k$, (ii) adapts them spatially via $\pi_k$, and (iii) keeps dynamics stable by reducing $\alpha_k$.

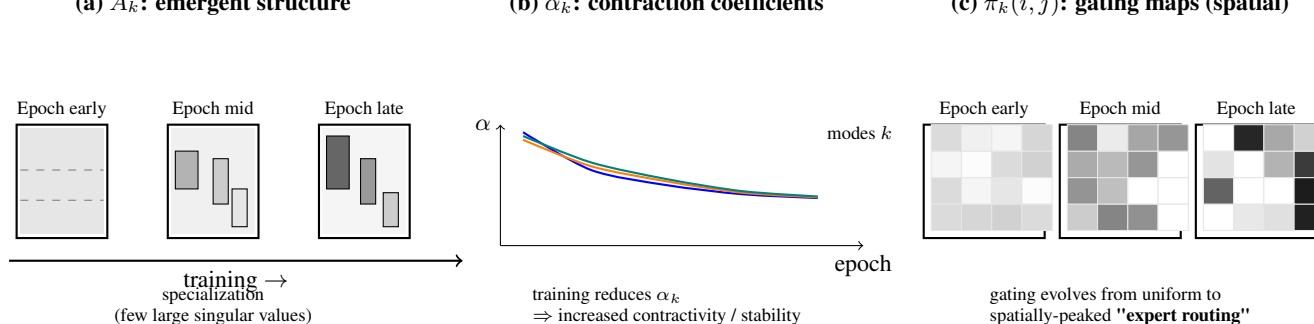

| (a) $A_k$: emergent structure | (b) $\alpha_k$: contraction coefficients | (c) $\pi_k(i, j)$: gating maps (spatial) |

Figure A.5: Schematic illustration of the typical evolution of LIFS parameters during training. (a) Example schematic of three snapshots of a learned affine map $A_k$: early (unstructured), mid (emerging directional blocks), and late (specialized dominant subspaces). (b) Typical trajectories of contraction coefficients $\alpha_k$ for several modes $k$: training progressively reduces $\alpha_k$, enforcing stronger contractivity and stabilizing the recursion. (c) Gating maps $\pi_k(i, j)$ at three training stages: from near-uniform to spatially-structured, peaked maps that implement local expert routing.

## B.4 Effect of the Number of Maps $K$ and Recursion Depth $D$

Increasing the number of contractive affine maps $K$ generally improves convergence smoothness and loss values. The optimal regime is typically $K \in \{4, 6\}$:

- $K = 2$ yields noticeable improvement but limited geometric diversity.

- $K = 4$ and $K = 6$ consistently produce stronger convergence curves.

- $K = 8$ shows diminishing returns and no visible degradation.

This confirms that a moderate number of affine maps provides enough geometric expressiveness while maintaining stability under spectral constraints. Deeper recursion improves both early and late convergence:

- For CIFAR datasets, performance saturates around ($D = 1$ or $D = 2$).

- For ImageNet-1K, deeper predictors ($D = 2$ or $D = 3$) continue improving.

- Depth $D = 4$ yields marginal gains and occasionally small slowdowns.

Larger $D$ strengthens the fixed-point refinement implicit in LIFS, but small images (CIFAR) saturate early due to limited spatial complexity.

**Faster Early Convergence.**   Across nearly all settings, JEPA+LIFS reduces the prediction loss more rapidly in the initial epochs. The contractive mixture updates provide multiple refinement steps per forward pass, enabling faster alignment with the target representation.

**Benefit of Recursion Depth $D$.**   Deeper recursion produces monotonic improvements in convergence and stability. CIFAR saturates at $D = 2$, (cf. Figure A.6) whereas ImageNet-1K continues to improve for $D = 3$ (cf. Figure 7). This highlights that the fixed-point iterative nature of LIFS is particularly suited for large-scale, high-resolution feature distributions.

**Overall View.**   The ablation trends confirm that LIFS is not merely a larger model, but an improved predictor class that replaces one-shot prediction with geometric recursive refinement. This explains why JEPA+LIFS exhibits both faster and more stable convergence across all benchmarks.

## B.5 General Discussion

This work proposes a shift in how predictive modules are designed within self-supervised architectures. Rather than interpreting the predictor as a shallow residual mapping, we frame it as a learnable dynamical system operating in latent space. This perspective exposes structural properties—such as contraction, specialization, and attractor formation—that are largely implicit or absent in conventional designs.

**From Predictors to Operators.**   Our formulation recasts the JEPA predictor as an explicit latent operator composed of multiple interacting affine transformations. By introducing an explicit operator structure, we gain direct control over spectral properties, enabling principled stability through contraction rather than reliance on architectural heuristics.

**Fractal Structure and Multi-Scale Modeling.**   The emergence of multiple specialized transformations suggests that JEPA+LIFS learns a structured family of latent updates rather than a single global mapping. This behavior aligns with classical interpretations of iterated function systems, where repeated application of contractive maps produces multi-scale attractors. In the context of representation learning, this provides a natural mechanism for capturing hierarchical and compositional structure without explicit supervision.

**Stability Beyond Optimization.**   While contraction directly improves optimization stability, its implications extend beyond training dynamics. Contractive operators induce robustness to perturbations in latent space, which may explain the observed resilience to noise and view variation. Moreover, the compatibility between contraction and EMA updates (cf. Corollary A.0.5), offers a principled explanation for why target networks remain stable without aggressive momentum tuning.

Figure A.6: **JEPA+LIFS vs. JEPA Baseline on CIFAR-100.** LIFS provides smoother and faster convergence than the baseline across all $(K, D)$ choices. The improvements saturate around $K = 4, 6$ and $D = 1, 2$.

**Relation to Prior Self-Supervised Methods.** Most prior work in self-supervised learning focuses on loss functions, augmentations, or encoder architectures. In contrast, our contribution targets the predictive pathway itself. This orthogonal axis of design suggests that gains in representation quality need not come from deeper encoders or stronger augmentations, but from enforcing structure in how latent predictions are formed.

**Limitations and Open Questions.** Although JEPA+LIFS introduces desirable stability properties, it also raises several open questions. First, the choice of the number of operators $K$ and their depth $L$ introduces additional hyperparameters. Second, while contraction aids convergence, overly restrictive constraints may limit expressivity in certain regimes. Finally, extending fractal operators to autoregressive or multi-modal settings remains an open direction.

**Broader Implications.** Viewing predictors as dynamical systems bridges self-supervised learning with control theory and nonlinear dynamics(cf. section: 6). This connection opens the door to importing tools such as Lyapunov analysis (Lohmiller & Slotine, 1998), attractor theory and stability certification into representation learning. We believe this perspective will become increasingly relevant as world models and long-horizon predictors grow in importance.

Table A.3: Hyperparameters used for different Encoders and databases.

| Parameter | SimpleCNNEncoder | ResNet-18 |
|---|---|---|
| Projector hidden dim | 512 or 1024 | 2048 |
| Predictor hidden dim | 512 or 1024 | 2048 |
| Number of maps $K$ | 6 | 6 |
| Recursion depth $L$ | 2 | 3 |
| Spectral threshold $\rho$ | 0.9 | 0.9 |
| $\lambda_{\text{var}}$ | 25 | 25 |
| $\lambda_{\text{cov}}$ | 1 | 1 |
| $\lambda_{\text{spec}}$ | 0.01 | 0.01 |
| $\lambda_{\text{div}}$ | 0.01 | 0.01 |
| Batch size | 128 | 128 or 256 |
| Warmup epochs | 10 | 10 |
| Learning rate | $10^{-3}$ | $10^{-3}$ |
| Weight Decay | $10^{-6}$ | $10^{-6}$ |
| EMA start | 0.99 | 0.99 |
| EMA end | 0.999 | 0.999 |
| Optimizer | Adam | Adam |
| Training epochs | 100 | 200 |
| Predictor embeding dim | 256 | 512 |

| Parameter | ViT-S/16 | ViT-B/16 |
|---|---|---|
| Projector hidden dim | 2048 | 2048 |
| Predictor hidden dim | 2048 | 2048 |
| Number of maps $K$ | 4-6 | 6 |
| Recursion depth $L$ | 2 | 3 |
| Spectral threshold $\rho$ | 0.9 | 0.9 |
| $\lambda_{\text{var}}$ | 25 | 25 |
| $\lambda_{\text{cov}}$ | 1 | 1 |
| $\lambda_{\text{spec}}$ | 0.01 | 0.01 |
| $\lambda_{\text{div}}$ | 0.01 | 0.01 |
| Batch size | 128 or 256 | 256 or 512 |
| Warmup epochs | 10 | 10 |
| Learning rate | $10^{-3}$ | $10^{-3}$ |
| Weight Decay | $10^{-6}$ | $10^{-6}$ |
| Optimizer | Adam | Adam |
| Training epochs | 100 | 200 or 300 |
| Predicted targets | 4 | 4 |
| Patch size | 16 | 16 |
| Predictor depth | 8 | 12 |
| Predictor attention heads | 6 | 12 |
| Predictor embedding dim | 384 | 768 |

