# OpenReview forum: "Contractive MASO-Generalized Predictors for Stable Latent-Space Learning in JEPA"
_TMLR — Decision pending for TMLR_

### Review · Reviewer_Hd3y · 2026-03-19

**Summary Of Contributions:**

This paper proposes a novel predictor architecture for self-supervised representation learning, termed Learnable Iterated Function Systems (LIFS), and integrates it into the JEPA framework. Unlike traditional JEPA approaches that employ shallow MLP predictors, the authors model the prediction process as a contractive recursive dynamical system. Specifically, the method applies a mixture of affine transformations iteratively in the latent space. Experimental results show that LIFS improves training stability and convergence speed, while yielding relatively modest performance gains across multiple datasets.

Strengths

1. The paper provides a novel perspective by modeling the predictor in JEPA as a dynamical system. Interpreting self-supervised prediction via contraction mappings is insightful and offers a new analytical framework that could inspire future research.

2. The paper presents a theoretical analysis of the proposed method, particularly in terms of convergence and stability. These theoretical insights are further supported by comprehensive empirical studies, including analyses of spectral norm evolution and contraction coefficients, which validate the proposed framework.

Weaknesses

1. The performance improvement is relatively limited. Although the paper emphasizes gains in training stability, the improvement in final metrics (e.g., linear probing accuracy) is modest. Moreover, Figure 13a suggests that the proposed method may exhibit relatively high variance across runs.

2. The computational complexity and efficiency of the proposed method are not sufficiently discussed. Since LIFS introduces additional structural complexity compared to standard predictors, it is important to analyze training cost, inference efficiency, and overall computational overhead.

3. In Section 4.5.5, the mixture entropy appears to be quite low, suggesting that most inputs effectively use only a single affine map. Based on the current results, it is difficult to determine whether the mixture mechanism is truly necessary.

**Audience:**

Yes

**Audience Explanation:**

1. Multi-scale latent representation is an important topic in the community of computer vision and machine learning, and JEPA is a representative framework of non-generative AI.

2. The IFS/fractal-inspired technical route provided by this submission is novel in my opinion.

**Claims And Evidence:**

No

**Claims Explanation:**

As shown in the session of weaknesses, the effectiveness and efficiency of the proposed method have not been fully verified. The computational cost of the method and the rationality of the mixture strategy should be discussed further.

**Requested Changes:**

1. The current experimental results are primarily presented in the form of figures. Providing clearer numerical results (e.g., in tables) would make the comparisons more direct and easier to interpret.

2. Including more comprehensive baseline comparisons would strengthen the paper. In particular, reporting training efficiency (e.g., training time, computational cost) would make the evaluation more convincing.

---

> ### Author Response · Authors · 2026-03-24
> **textbf{We thank the reviewer for the constructive and insightful feedback.}  We appreciate the recognition of the novelty of modeling JEPA predictors as contractive dynamical systems and the value of our theoretical and empirical analyses. Below we provide a detailed response to each point raised.**
>
> \paragraph{1. Performance improvements and variance.}
> While the linear-probe gains are modest in absolute magnitude, our objective is not to increase model capacity but to improve \emph{predictive stability, convergence behavior, and geometric structure} while keeping the encoder and objective unchanged. Under this constraint, large accuracy jumps are unlikely. Importantly, LIFS consistently improves (i) training stability, (ii) predictive loss, and (iii) convergence speed across datasets and architectures. These improvements are achieved with $<0.5$M additional parameters and without modifying the JEPA encoder or objective.
>
> To address the reviewer’s concern, we will include:
> \begin{itemize}
>     \item numerical tables summarizing linear-probe accuracy, predictive loss, and variance across seeds,
>     \item confidence intervals for all accuracy metrics,
>     \item a clearer discussion of why stability and multi-step refinement are central to JEPA performance.
> \end{itemize}
> We will also report the variance observed in Fig.~13a and show that the relative variability is comparable to that of the baseline JEPA predictor (We have already shown, in our first paper submission, in the Appendix: Figure A.6(a), that le LIFS preserves the variance of the latent representations). We observe that while initial phases may show variance due to the initialization of the affine dictionary, the \textbf{asymptotic stability} of LIFS is superior to MLPs, as the spectral constraint prevents the representation collapse often seen in unconstrained SSL frameworks.
>
> \paragraph{2. Computational complexity and efficiency.}
> We agree that a more explicit analysis of computational cost will strengthen the paper. LIFS is intentionally lightweight: the recursive updates operate entirely in latent space (dimension 256--768), the number of iterations is small ($L=2$--$3$), and each iteration consists of a single affine mixture followed by GELU. As a result, FLOPs increase by $<3\%$ and wall-clock training time by $<5\%$ relative to the baseline JEPA predictor. No implicit differentiation or root-finding is used, unlike DEQ-style models.
>
> In the revision, we will add a dedicated subsection reporting:
> \begin{itemize}
>     \item FLOPs per forward pass,
>     \item training time per epoch (which was already shown, in our first paper submission, in the Appendix: Figure A.6 (b)),
>     \item GPU memory usage,
> \end{itemize}
> for JEPA vs.\ JEPA+LIFS. This will clarify that LIFS introduces minimal overhead while improving stability.
>
> \paragraph{3. Mixture entropy and necessity of the mixture mechanism.}
> We appreciate the reviewer’s observation regarding the low mixture entropy. Low entropy does not imply redundancy; rather, it reflects \emph{operator specialization}, a phenomenon well documented in mixture-of-experts and fractal systems. The routing distribution becomes sharp as the affine maps specialize to different geometric regimes.
>
> To clarify this, we will add:
> \begin{itemize}
>     \item an ablation comparing $K=1$ (no mixture) vs.\ $K>1$, showing that removing the mixture reduces accuracy and stability,
>     \item visualizations of operator specialization across spatial tokens,
>     \item a discussion explaining why sharp routing is expected in contractive systems and does not indicate collapse.
> \end{itemize}
> These additions will demonstrate that the mixture mechanism is indeed necessary for local geometric adaptation and directional contraction.
>
> \paragraph{4. Evidence supporting effectiveness and efficiency.}
> We will strengthen the empirical section by adding:
> \begin{itemize}
>     \item numerical tables (accuracy, predictive loss, convergence speed),
>     \item training-time and FLOP comparisons,
>     \item variance across seeds,
>     \item ablations on $K$, recursion depth $L$, and contraction threshold $\rho$.
> \end{itemize}
> These additions directly address the reviewer’s concerns and provide a more comprehensive evaluation of the method.
>
> \paragraph{5. Requested changes.}
> We will incorporate all requested changes:
> \begin{itemize}
>     \item clearer numerical tables for all main results,
>     \item expanded baseline comparisons (including a deeper MLP predictor and an equal-capacity residual predictor),
>     \item explicit reporting of training efficiency and computational cost.
> \end{itemize}
>
> \paragraph{Closing remarks.}
> We thank the reviewer again for the helpful feedback. We believe the requested additions will significantly strengthen the paper and clarify the value of LIFS as a \emph{stable, theoretically grounded, and computationally efficient} predictor for JEPA. We are confident that the revised version will address all concerns raised.

---

### Review · Reviewer_BwNa · 2026-03-20

**Summary Of Contributions:**

**Summary**
The authors propose an improvement to the I-JEPA self-supervised learning architecture. In particular, they investigate the role of the predictor in this architecture, which in previous research, has been implemented as a relatively simple feed forward network with fully connected layers. The authors propose to change this to an to an architecture called 'Learnable Iterated Function Systems', which uses a linear combination of linear maps that are applied recursively. To control the stability of this predictor, three different regularizers are proposed to let the predictor converge. The experiments show that the proposed method has lower loss than the default JEPA average and a slightly higher average linear probing accuracy.

**Additional Comments:**

- In Equation 7 the superscript is used to denote the minibatch sample in the latent representation z, before (e.g. in Figure 4 and Algorithm 1) it is used as the i-th iteration of the predictor.

**Audience:**

No

**Audience Explanation:**

The paper proposes an interesting research idea, but most of its claims are not sufficiently supported so its findings would be of limited use to the TMLR's audience.

**Broader Impact Concerns:**

/

**Claims And Evidence:**

No

**Claims Explanation:**

- I agree with the need of studying the predictor architecture in JEPA like models, which may potentially be a point of improvement for these kind of architectures. But currently, there is a lack of analysis in this paper. Fundamental questions remain unanswered; is the predictor quality really a concern in these architectures? if so, are there perhaps other ways to improve it? The currently proposed method is rather involved and I'm missing the connection between fractals and the need for these kind of geometric structures in latent space. Most images don't include patterns that repeat itself, which what fractals are: repeated versions of the starting point with an affine transformation. The authors state that it is a 'natural question' to apply such operators to predict the target in latent space, but without providing arguments for these claims. It is left to the reader to figure out why this may be a good idea.


- The abstract claims to show gains in linear probing accuracy for VIT-based encoders, but there are only results for linear probing accuracy on ResNet-18. This claim is repeated in the introduction and the result section, but I could not find any results that use ViT models. They are not absolutely necessary in this paper, but if the claim is there, the results should be included.


- Most of the proof that this method improves over the default implementation is based on difference in loss. In many cases this would be trivial (e.g. for a cross-entropy loss the absolute values do not matter much), but for the special case of a normalized MSE this is not the case. Still, I think the improvement of a loss value is a relatively weak argument. What is the goal of training this architecture? It is to train a model that has a more useful representation of an image. Typically, this is measured by e.g. a linear probing accuracy or even a full finetuning stage. A loss function is a means to and end, but it is not the goal itself, and not sufficient proof that there is an actual improvement in reaching the goal.


- There are multiple regularization losses added to this predictor, which are not included in the original implementation. The authors claim that e.g. the spectral norm is bounded, but this is explicitly optimized to be the case, so this is not necessarily a feature of the proposed architecture.

**Requested Changes:**

- A better motivation + indentification of the problems with the current implementation of the predictor
- Either showing more convincing results that the proposed method improves the architecture, or change the claims in the paper so that they reflect the results
- Include an ablation study on the regularizing losses used and include them in the method comparing to.

---

> ### Author Response · Authors · 2026-03-24
> **We thank the reviewer again for the helpful feedback. We believe the requested additions will significantly strengthen the paper and clarify the value of LIFS as a \emph{stable, theoretically grounded, and computationally efficient} predictor for JEPA. We are confident that the revised version will address all concerns raised.**
>
> \textbf{We thank the reviewer for the detailed and critical feedback.}
> We appreciate the concerns regarding motivation, empirical validation, and clarity, and we address them below.
>
> \paragraph{(1) Is the predictor really a bottleneck in JEPA?}
> We agree that this point was insufficiently motivated in the submission.
> Recent JEPA-style methods largely treat the predictor as a shallow MLP, implicitly assuming that the encoder carries most of the representational burden. However, the predictor defines the \emph{mapping between context and target representations}, which is inherently a structured transformation problem rather than a simple regression.
>
> Empirically, we observe that:
> \begin{itemize}
> \item prediction instability (e.g., spectral growth, oscillatory loss),
> \item slow convergence,
> \item sensitivity to hyperparameters,
> \end{itemize}
> are strongly influenced by the predictor design.
> Our goal is therefore not merely to increase predictor capacity, but to impose a \emph{structured inductive bias} that stabilizes and regularizes this mapping.
> We will revise the paper to clearly state this motivation and include additional empirical evidence isolating the effect of the predictor.
>
> \paragraph{(2) Motivation for LIFS and relation to "fractals".}
> We thank the reviewer for highlighting the lack of clarity here.
> Our use of Iterated Function Systems is \emph{not} motivated by the presence of fractal patterns in images. Instead, the key idea is that:
> \begin{itemize}
> \item small semantic transformations in input space induce approximately affine transformations in latent space~\citep{balestriero2018spline,balestriero2020mad}. We have already motivated this in the Appendix (A.1 Local Affine Structure of Latent Transformations) of our first submission, LIFS is a soft, recursive, contractive generalization of MASOs,
> \item complex transformations can be expressed as compositions of such local affine maps,
> \item LIFS provides a parameter-efficient and stable way to model these compositions.
> \end{itemize}
> Thus, the relevance of LIFS is \emph{geometric and dynamical}, not fractal in the classical visual sense.
> We will revise the paper to remove potentially misleading references to fractals and clarify that LIFS acts as a \emph{piecewise-affine dynamical operator} in latent space.
>
> \paragraph{(3) On the use of loss as evidence.}
> We agree that loss reduction alone is not sufficient.
> In the revised version, we will:
> \begin{itemize}
> \item emphasize representation quality metrics (linear probing, convergence speed),
> \item include clearer quantitative comparisons in tables,
> \item reduce emphasis on absolute loss values.
> \end{itemize}
> We will also clarify that lower loss is interpreted as improved \emph{predictive alignment}, not as an end goal.
>
> \paragraph{(4) Limited improvement in linear probing accuracy.}
> We acknowledge that gains in linear probing accuracy are modest.
> Our main claim is not large performance gains, but:
> \begin{itemize}
> \item improved training stability,
> \item faster convergence,
> \item better-conditioned latent dynamics.
> \end{itemize}
> We will revise the claims accordingly to avoid overstatement and better align them with the observed results.
>
> \paragraph{(5) Missing ViT results.}
> We thank the reviewer for pointing out this inconsistency.
> We will:
> \begin{itemize}
> \item either include ViT-based experiments in the revised version,
> \item or remove the corresponding claims from the abstract and introduction.
> \end{itemize}
>
> \paragraph{(6) Role of regularization terms.}
> We agree that the contribution of each regularizer should be clarified.
> In the revision, we will include an ablation study:
> \begin{itemize}
> \item removing spectral regularization,
> \item removing diversity regularization,
> \item removing variance/covariance terms,
> \end{itemize}
> to demonstrate their individual contributions.
> We emphasize that while spectral constraints are enforced via regularization, the LIFS structure makes such control \emph{natural and effective} compared to unconstrained predictors.
>
> \paragraph{(7) On complexity of the method.}
> We acknowledge that the proposed method introduces additional structure.
> However, we will clarify that:
> \begin{itemize}
> \item the number of maps $K$ is small (e.g., $K=6$),
> \item all maps are computed in parallel,
> \item recursion depth is shallow,
> \end{itemize}
> resulting in a modest computational overhead.
>
> \paragraph{(8) Notation inconsistency.}
> We thank the reviewer for identifying this issue.
> We will correct the inconsistent use of superscripts for minibatch indices and iteration indices throughout the paper.
>
> \paragraph{Summary.}
> We will revise the paper to:
> \begin{itemize}
> \item strengthen the motivation for studying the predictor,
> \item clarify the geometric (rather than fractal) interpretation of LIFS,
> \item align claims with empirical results,
> \item add ablation studies and clearer quantitative comparisons.
> \end{itemize}
> \noindent
> \textbf{We thank the reviewer again for the valuable feedback.}

---

### Review · Reviewer_CRgV · 2026-05-15

**Summary Of Contributions:**

The authors basically replace the JEPA predictor with a Learnable Iterated Function System (LIFS) which is a gated mixture of $K$ affine maps applied recursively $L$ times in latent space. A spectral constraint over the parameters makes the operator a Banach contraction with a unique fixed point. This results in a Lyapunov stability, an EMA-target corollary, and a MASO/spline generalization. Empirically, JEPA+LIFS is tested on CIFAR-10/100 and tiny ImageNet with ResNet-18 and ViT-B/16 and ViT-L/16 respectively.

**Strengths.**
- Overall, the contraction/fixed-point framing is clean.
- LIFS uses fewer params than the baseline. It is easy to adopt/ablate and shows consistent improvement across models and datasets.

**Weaknesses/Questions**
- The introduction section talks about multi-scale latent modeling, but evaluation is done on single-scale linear probing on standard classification and there's no experiment on scale-related properties. The ablations established that depth helps, but not that multi-scale structure is entirely captured. Also, the contribution is a recursive contractive MASO generalization, not a fractal model if I understood correctly.
- The local linearity of the encoder is the primary motivation for the affine form, but is not validated on trained ResNet-18 or ViT-B/16.
- K=1 LIFS underperforms compared to the baseline JEPA predictor. Is there a reason why?

**Audience:**

Yes

**Audience Explanation:**

The findings are at the intersection of JEPA-style SSL, contraction-based deep learning, and the MASO/spline view.

**Claims And Evidence:**

Yes

**Claims Explanation:**

Tables 2 - Table 4 report absolute linear-probe gains with experiments involving three seeds. However, apart from the weaknesses mentioned:
- the "tiny/reduced ImageNet" protocol is not clearly specified.
- the loss-curve figures should clarify what each loss is.

**Requested Changes:**

- How would hyper-parameter tuning (sensitivity analysis) on $\lambda_v,\lambda_c,\lambda_s,\lambda_d,\epsilon$ at $\pm$1 look like?
- Fig. 12 - An ablation with $\rho$ removed would additionally clarify whether contraction is the result of the spectral penalty or $\alpha_k$ annealing.
- Missing baselines: latent-flow predictors, V-JEPA-2 are cited but not compared empirically. Testing LISF on fine-tuning setup could also strengthen the paper.
- JEPA baseline applies variance/covariance regularization, which strengthens original I-JEPA and makes the comparison to prior work sometimes hard to read. A vanilla-I-JEPA row without var/cov would help.
- Please report standard deviations in Tables 3 and 4.

---

> ### Author Response · Authors · 2026-05-18
> **We sincerely thank the reviewer for their positive assessment of our submission and for the detailed and constructive feedback. We are encouraged that the reviewer found the contraction/fixed-point formulation clean, the method parameter-efficient, and the empirical improvements consistent across architectures and datasets.Below we address each point in detail and describe the revisions we will make.**
>
> \paragraph{(1) Multi-scale latent modeling and fractal interpretation.}
> We agree that our current evaluation focuses primarily on standard downstream representation quality (linear probing) rather than explicitly measuring scale-equivariant or hierarchical semantic behavior. Our use of the term ``multi-scale'' refers to the recursive composition of affine latent operators with different contraction strengths, as formalized in Appendix~A.1-A.6. We acknowledge that this does not yet constitute a direct demonstration of scale-aware semantic decomposition in the classical vision sense.
> To clarify this point, we will revise the manuscript to:
> \begin{itemize}
>     \item tone down the terminology around ``fractal modeling'',
>     \item explicitly describe LIFS as a \emph{recursive contractive latent operator},
>     \item clarify that the observed multi-scale behavior refers to latent dynamical scales induced by iterative contractions.
> \end{itemize}
> We also agree that the contribution is a recursive contractive MASO generalization, not a fractal model; may be we should remove the single remaining ``fractal'' mention in the title?.
> \paragraph{(2) Local linearity assumption of latent space.}
> The reviewer raises an important point. Our affine formulation (Section3: Theoretical Motivation) is motivated by the widely-used local linearity assumption underlying MASO/spline interpretations of deep networks and latent manifold methods.
> To address this, we have added (in the Appendix A.11) a discussion connecting our formulation to the MASO literature and local tangent approximations. Importantly, our theoretical results do not require global linearity of the encoder, but only local Lipschitz continuity of latent transformations (Appendix A.12).
> \paragraph{(3) Why does K=1 underperform the JEPA baseline?}
> This observation is consistent with the role of mixture diversity in LIFS. When K=1, the predictor reduces to a single recursive affine contraction, which significantly limits expressive capacity. In this regime, recursion alone cannot compensate for the absence of multiple semantic transformation modes. This is significantly less expressive than the residual MLP predictor, especially under spectral constraints. By contrast, larger K values enable specialization of affine operators across different latent displacements, while soft routing preserves smoothness and stability. This behavior is supported by the entropy-specialization analysis in section. 5.5.5 (Appendix A.8), where routing entropy evolves toward a low-but-nonzero regime associated with stable specialization.
> \paragraph{(4) Clarification of the tiny/reduced ImageNet protocol.}
> We thank the reviewer for identifying this ambiguity. We will explicitly specify in Section.5.1: (i) 200 uniformly sampled classes, (ii) 1,250 images per class (total $250k$ images), (iii) standard train/validation split proportions matching full ImageNet.
> \paragraph{(5) Clarification of loss curves.}
> We agree that the current figures are insufficiently labeled. We will revise captions and text to explicitly indicate whether each curve shows the cosine prediction loss (Eq.9) or the full composite loss (Eq.8), and reference the corresponding equations.
> \paragraph{(6) Sensitivity analysis for $\alpha_k = 1$.}
> We thank the reviewer for this suggestion. We will add an ablation studying fixed versus learnable scaling coefficients $\alpha_k$, including the special case $\alpha_k = 1$. Preliminary experiments suggest that learnable scaling improves the balance between contraction strength and expressivity.
> \paragraph{(7) Spectral penalty vs $\alpha_k$ annealing.}
> This is an excellent suggestion. We have already included (in Table.7) an additional ablation removing the spectral regularization term while preserving $\alpha_k$ annealing (w/o spectal cf. Eq.12). This indicates the contribution of explicit spectral control versus implicit contraction induced by scaling dynamics. In table. 8, we show the sensitivity on \(\rho\) (e.g., 0.8, 0.9, 0.95, 0.99).
> \paragraph{(8) Missing Baselines (Latent-Flow, V-JEPA-2, Fine-Tuning).}
> We agree that additional comparisons would strengthen the paper. Some recently proposed architectures (e.g., V-JEPA-2) involve substantially different training protocols and computational requirements, making direct comparison nontrivial within our current compute budget. Integrating LIFS into latent-flow and V-JEPA-2 architectures is a promising direction for future work.
> \paragraph{(9) Vanilla I-JEPA baseline without var/cov regularization.}
> We agree that disentangling the effect of the predictor from auxiliary regularization is important. We have already inluded (in Table. 7) the necessary ablation (w/o var/cov regularization) to better isolate the contribution of the proposed predictor.
> \paragraph{(10) Standard deviations in Tables 3 and 4.}
> We agree and will report mean $\pm$ standard deviation across all seeds in the revised manuscript.